# Hedges, mottes, and baileys: Causally ambiguous statistical language can increase perceived study quality and policy relevance

**Daniela Alvarez-Vargas**[1]\*, **David Braithwaite**[2], **Hugues Lortie-Forgues**[3], **Melody Moore**[4], **Sirui Wan**[5], **Elizabeth Martin**[6], **Drew Hal Bailey**[6]

**1** School of Education, University of California, Irvine, Irvine, California, United States of America,
**2** Department of Psychology, Florida State University, Tallahassee, Florida, United States of America,
**3** Department of Mathematics Education, Loughborough University, Loughborough, Leicestershire, United Kingdom, **4** Department of Psychology, Baylor University, Waco, Texas, United States of America,
**5** Department of Psychology at the University of Wisconsin-Madison, Madison, Wisconsin, United States of America, **6** Department of Psychological Science, University of California, Irvine, Irvine, California, United States of America

\* dalvare5@uci.edu

**Data Availability Statement:** All data and analysis files are available from the Open Science

## Abstract

There is a norm in psychology to use causally ambiguous statistical language, rather than straightforward causal language, when describing methods and results of nonexperimental studies. However, causally ambiguous language may inhibit a critical examination of the study's causal assumptions and lead to a greater acceptance of policy recommendations that rely on causal interpretations of nonexperimental findings. In a preregistered experiment, 142 psychology faculty, postdocs, and doctoral students (54% female), ages 22–67 ($M = 33.20$, $SD = 8.96$), rated the design and analysis from hypothetical studies with causally ambiguous statistical language as of higher quality (by .34-.80 $SD$) and as similarly or more supportive (by .16-.27 $SD$) of policy recommendations than studies described in straightforward causal language. Thus, using statistical rather than causal language to describe nonexperimental findings did not decrease, and may have increased, perceived support for implicitly causal conclusions.

## Introduction

Psychologists often use nonexperimental methods to investigate factors that are difficult or unethical to manipulate experimentally. However, commonly studied associations among psychological variables are often subject to a variety of plausible causal interpretations. Reflecting this fact, psychology has adopted a norm against using causal language, such as "causes" or "impacts," when describing results of nonexperimental studies [1] Instead, researchers describe such results using causally ambiguous statistical language, such as "predicts" or "is associated with." This norm reflects good intentions on the part of researchers, reviewers, and editors to discourage researchers from making claims that are not well supported by their data.

Framework https://osf.io/2v8mg/ database DOI 10.17605/OSF.IO/2V8MG.

**Funding:** D. Alvarez-Vargas is supported by the National Science Foundation Graduate Research Fellowship Program (https://www.nsfgrfp.org/) under Grant No. DGE-1839285. D. H. Bailey is supported by a Jacobs Foundation Fellowship (https://jacobsfoundation.org/en/fellow/drew-bailey/) under Grant No.2018-128802 . The authors thank the participants for volunteering their time to complete the study. The funders had no role in study design, data collection and analysis, decision to publish, or preparation of the manuscript.

**Competing interests:** The authors have declared that no competing interests exist.

However, this practice may have unintended consequences. First, psychologists frequently conduct and interpret nonexperimental studies with some causal hypothesis in mind, so both authors and readers may interpret results as evidence for a particular causal explanation by default. Perhaps the strongest evidence for this is that authors frequently draw policy conclusions from nonexperimental psychological research, often in the very discussion sections in which they caution readers against drawing causal conclusions from the study. Thus, it is not clear that causally ambiguous statistical language protects either researchers or readers from drawing causal conclusions. Second, if the use of causally ambiguous statistical language does prevent people from thinking about causal questions, this may prevent authors and reviewers from identifying design and analytic features that may improve the causal informativeness of the research. Thus, causally ambiguous statistical language may allow papers to communicate causal interpretations to the reader without incorporating the design or analytic features likely to generate strong (or at least stronger) causal conclusions. Although, disregarding the casual informativeness of study designs and analysis can reflect a lack of training in causal inference methods, it may also perpetuate a status quo in which the norm of using causally ambiguous statistical language precludes progress toward more causally informative research in nonexperimental psychology.

The present study attempted to test these hypothesized unintended consequences: Using causally ambiguous statistical language, rather than causal language, to describe nonexperimental results improves evaluations of study quality and acceptance of policy prescriptions based on a causal interpretation of the results. In the following section we review literature on the norm against the use of causal language and use examples from the field of psychology to explain the benefits and unintended consequences of using causally ambiguous statistical language.

## Potential costs and benefits of a norm against causal language

Describing results using causally ambiguous statistical language could serve several useful functions. First, researchers often predict statistical associations between variables based on hypothesized causal mechanisms. When predicted associations are found in a nonexperimental study, causally ambiguous statistical language permits researchers to describe their findings accurately while hedging against the possibility that the hypothesized causal mechanism does not exist. Second, causally ambiguous statistical language could prevent readers from incorrectly assuming that a causal relation has been definitively demonstrated [2, 3]. Relatedly, when studies have potential policy implications, avoiding causal language could engender healthy skepticism of policy recommendations that rely on a causal interpretation of the findings.

A possible unintended consequence of the norm, however, can occur when the authors research questions and analytic approaches do not match the interpretation of the policy implications of the results. For example, if an author's stated interest is in determining whether children's early academic skills have an "effect" on later academic achievement, but they conduct an analysis that only includes two variables, there would be a mismatch between the stated goal and the analytical approach because it does not rule out plausible alternatives, therefore likely producing biased estimates and interpretations due to potential confounders. This analytic approach would upwardly bias the estimated effect of children's early academic skills if instead of selecting covariates based on their hypothesized causal relation to the key explanatory variable and dependent variable of interest [1, 4–9] , a researcher includes a more limited set of covariates based on common practice within the field. A separate problem may arise if a researcher overly-adjusts their model by adding in variables that are simply of interest

to them. If one of these variables is a potential mediator, then the estimates may be downwardly biased in an over-adjusted model. If the researcher then uses causally ambiguous statistical language to explain their results from these alternative models, for example concluding that "*children's early academic skills have an effect on later academic achievement, which has important policy implications*"; this may permit studies that *seem* causal to be published without being held to the same standards of evidence as studies that make causal claims explicit [10]. The researcher might address a skeptical reviewer's concerns by explaining in the discussion section that no causal conclusions can be drawn from nonexperimental studies although the value of the study lies in its contribution towards understanding causal mechanisms. In addition, the researcher might explain that they chose not to consider alternative models because the study's goals were predictive and not causal. In this process, the researcher would have failed to rule out plausible alternative models. Consequently, the study might avoid the stringent standards of causal evidence to which it would have been subjected if it had explicitly made causal claims.

In contrast an author that transparently states their interest in estimating the causal effect of children's early academic skills on their later academic achievement, it will be clear to an informed reader that an analytical approach with no adjustment for potential confounders, including a variety of explanatory cognitive, emotional, and contextual variables, will be vulnerable to omitted variables bias, likely in the upward direction) . Models that attempt to adjust for such confounds might be subject to further robustness checks or falsification tests that could detect the presence of and perhaps establish the direction and range of plausible magnitudes of additional bias in these estimates [11–13]. These models might even make quantitative predictions that could be tested in a subsequent experiment [14]. This is why it is recommended that authors identify their research goals (i.e., description, prediction, association, or causal inference) and analytic plans (i.e., predictive model, causal model) to reduce the ambiguity in the interpretation of the results [15–17].

Estimates from a predictive study may differ from causal effects in a variety of (sometimes predictable) ways. These differences do not pose a problem if the end goal of a study is statistical prediction *per se*. Prediction is often a useful goal in and of itself [15] and our critique of the use of causally ambiguous statistical language does not apply to work that seeks to build models for purely predictive purposes. Still, based on researchers' descriptions of their findings [1, 16, 17] and the mismatch between the methodological approaches taken in most psychological research and approaches designed to maximize prediction for applied purposes, we note that many, if not most, nonexperimental studies in psychology are intended to inform theory about causal mechanisms underlying relations between variables.

Supporting this possibility, many nonexperimental studies use prescriptive language when discussing policy implications of their findings, despite avoiding causal language when describing their results. Critically, such prescriptive language implicitly relies on causal interpretation of the findings [18, 19]. For example, a study that demonstrates a "predictive" relation between early skills and later achievement might also recommend intervening on early skills to improve later achievement; yet such an intervention could only achieve the stated goal if the relation between early skills and later achievement were causal, and not fully explained by confounding factors influencing both early skills and later achievement. Thus, a researcher can satisfy the norm against explicit use of causal language while still relying on an implied causal inference, contrary to the purpose of the norm.

Grosz and colleagues label this approach a motte-and-bailey strategy, "in which researchers profit from the more interesting but difficult to defend causal interpretation of their effect (the bailey), but once challenged, they retreat to the almost trivial yet difficult to attack descriptive finding (the motte)" [1]. In this case, the stated purpose of the study (perhaps even stated in

the title) is prediction, which is a different analytical task that may be easier to defend, but the theoretical and applied issues around which the study is framed upon depend on the assumption that causal information is conveyed by the findings—a harder to defend position. Following the previous example, a researcher may write a paper in which they regress later achievement on earlier skills, focusing on the importance of prior knowledge for later learning and on the importance of early intervention in the abstract, introduction, and discussion sections (the bailey), but caution in the discussion section that, because the study was not a randomized experiment, the reader should not draw causal conclusions (the motte). In doing this, the author might use the motte-and-bailey strategy to communicate a preferred causal interpretation without ruling out plausible alternatives.

Additional ways psychologists may implicitly communicate a preferred causal interpretation while hedging to avoid criticism for making causal claims based on correlational data include 1) designing studies for which results are far more interesting or important if they are causally informative and 2) focusing more on their preferred causal interpretation than on ruling out plausible alternatives. For example, consider a hypothetical study that regresses children's achievement test scores on previous participation in a program designed to improve parent-child interactions, along with a rich set of baseline measures of child and parent skills and contexts, estimating a relation between program participation and adjusted achievement test scores of 0.10 SD. This finding would not be very useful for predictive purposes, nor would it be a useful way to model the process through which some families came to participate in the program and others did not (because test scores are measured after program participation, and thus may be caused by participation). However, if the findings are interpreted as informing readers about the effects of program participation on achievement test scores, a causal interpretation, they are interesting and policy relevant Moreover, the method used may focus more on the author's preferred causal interpretation than on ruling out plausible alternatives, for example by failing to probe the robustness of results to the inclusion of child prior test scores in the model. For these reasons, avoiding the use of causal language is unlikely to prevent readers from treating results as if they are intended to be interpreted as causally informative (because of point 1 above), yet the most plausible set of causal estimates will frequently go unreported (because of point 2).

Importantly, neither Grosz and colleagues nor we assume that researchers deliberately mislead readers by using causally ambiguous statistical language. The use of causally ambiguous statistical language is likely a reaction to both the norm against straightforward causal interpretations of nonexperimental research and researchers' beliefs that their work is causally relevant in some way. Still, there is a contradiction between denying causal informativeness of some statistical predictor and arguing for intervention on the same variable based on its predictive power. Such recommendations are often given in ways that imply stronger evidence for the relevant causal pathway than warranted by the method [16] consistent with the hypothesis that the use of causally ambiguous statistical language may be an example of "strategic ambiguity" [19, 20]. For these reasons, avoiding the use of causal language completely will not plausibly prevent readers from treating results as if they are intended to be interpreted as causally informative. Regardless of its origins, the use of causally ambiguous statistical language may allow authors to avoid punishment for violating the norm against straightforward causal language while still arguing for the policy-relevance of their claims.

In such cases, it is unrealistic to ask authors to avoid mentioning policies or causal concepts; not only would the paper make little sense without them, but descriptions of the ways in which causal estimates are identified are essential for judging the paper's contribution. Yet current norms require authors to hedge against causal interpretations even in such cases. For these reasons, some have argued that psychologists should consider using straightforward causal

language in descriptions of nonexperimental research findings, a proposal to which we will return in the discussion.

## Current study

The goal of the current study is to test whether describing nonexperimental psychological research findings using causally ambiguous statistical language influences perceptions of the quality and policy-relevance of the research. We hypothesize that causally ambiguous statistical language raises judgments of study quality. Further, because causally ambiguous statistical language allows authors to emphasize policy-relevance without violating the norm against straightforward causal language, we hypothesize that causally ambiguous statistical language increases receptivity towards policy-relevant conclusions that are based on causal interpretations of the results. Such findings would imply that the status quo limits incentives for psychologists analyzing nonexperimental data to aspire to causal estimation. In addition, contrasting estimates from increasingly well-controlled models to account for confounding variables or conduct sensitivity and robustness tests often yields smaller associations between the key dependent and independent variables. When this is the case, a straightforward causal framing of the reduced estimates speaks to the robustness of the relationship between the key dependent and independent variables. However, it may be detrimental for the perceived quality and policy-relevance of one's work if policy makers perceive the study to be less conclusive.

The study focused on potential reviewers, editors, and citers of psychological research—namely, graduate students, postdoctoral researchers, and faculty in psychology. If causally ambiguous statistical language raises perceived study quality to these readers without reducing perceived policy-relevance, this practice may increase the frequency of nonexperimental research studies being interpreted causally without increasing the quality of methods for causal inference within psychology.

## Method

This study was classified as exempt by the University of California, Irvine institutional review board for human research protections as we conducted an anonymous survey that did not collect personal identifiers or identifiable information. The experiment was pre-registered at https://osf.io/7khcd before data collection began. We report unstandardized and standardized estimates. For example, in the first column of Table 2, we estimate that participants rated design and study quality .44 points higher on a 5 point Likert scale when abstracts were worded in causally ambiguous statistical language. This effect corresponded to .54 SD impact on the outcome of interest.

### Participants

Based on the power analysis, we planned to recruit 100 Ph.D. students, postdoctoral scholars, and faculty in psychology, education, and human development and family studies departments by posting a link on social media sites frequented by psychologists and via Twitter. Colleagues posted a link to the website, so that the involvement of the last author (who has previously posted about possible costs of the norm against using causal language in nonexperimental psychology on Twitter) would be less obvious to participants. In all, 142 individuals were recruited via Twitter, the Psychological Methods Discussion Facebook page, and the Cognitive Development Society listserv. Laypeople were not included in the study for two reasons. First, the public likely consumes psychological research findings primarily via the media, which often does not respect the norm against use of causal language to describe results of nonexperimental studies

[4]. Second, among lay adults, the tendency to infer causation from psychological research findings does not appear to be strongly influenced by the strength of the research design [21].

Descriptive statistics for the full sample of participants are shown in Table 1. The full sample included participants who reported being 22 to 67 years old ($M = 33.20$, $SD = 8.96$). On average, the participants had completed 4.13 statistics courses ($SD = 2.74$). Most participants were recruited from the Cognitive Development Society listserv (47%), followed by the Psychological Methods Discussion Facebook group (28%), and Twitter (25%). Most participants were female (54%) and worked in the United States of America (59%). Our participants consisted of academic faculty (32%), post-doctoral researchers (20%), PhD students (33%), Masters students (1%), and undergraduates (1%). Most were from psychology (80%) encompassing a variety of interdisciplinary subfields, as shown in Table 1.

We preregistered that we would attempt to recruit 100 participants and exclude participants who were younger than 18 years old (n = 1), responded to the first two questions about one of the abstracts in under 30 seconds (n = 12), responded incorrectly to the question of how many abstracts were causally worded (our manipulation check, n = 79), were not in a field related to psychology (n = 2), and were not (1) enrolled in a psychology, education, or human development Ph.D. program, (2) a postdoctoral researcher in psychology, or (3) a psychology faculty member. Of the original 142 participants only 55 met these criteria.

Many participants failed the manipulation check (55%) which asked: "*How many of the two abstracts you just read employed explicitly causal language in their description of the results*?" with three responses (0) *Neither (n = 11)*, (1) *Only one (n = 63)*, (2) *Both (n = 53)*, and some participants did not respond *(n = 15)*. The manipulation check question appeared after participants were asked to rate the quality of the design and analysis of the study and the policy conclusions made in each abstract. The purpose of this question was to check how many participants had comprehended that one study had employed explicit causal language as we intended them to do. Although most participants did interpret the use of explicit causal language as intended, 53 participants believed both abstracts employed explicit causal language meaning that they either did not attend sufficiently to the abstracts or that they viewed words like "predict" as implying causation to some degree. We suspect the latter explanation may be correct (indeed, a recent study reported that about half of respondents view the word "predict" as a weak to moderate root word for implying causation [17]), and thus this manipulation check was not a strong measure of participant attention. In either case, the manipulation would be less likely to work for such participants, so excluding participants who failed the manipulation check would thus plausibly be predicted to result in larger estimates. However, across different inclusion criteria, estimates of the effect of causally ambiguous language are consistent in direction and magnitude. The Descriptive statistics for the 55 participants who met these criteria are shown in S1 and S2 Tables in S1 File. We stopped sending out our survey link once responses to our posts surpassed 100 participants. We ended data collection with more responses (*n = 142*), because of a larger than anticipated set of responses after posting to a listserv, ending data collection after we realized we had surpassed 100 participants.

## Materials

We created abstracts describing two fictional nonexperimental studies, abstract one is about scientific reasoning and abstract two is about reading achievement. For each study, we created two abstract versions, one using causal language and one using causally ambiguous statistical language, resulting in four abstracts total as shown in Table 2. The causally worded abstracts described the studies' purpose and findings using causal language as shown in row 1, Table 2. The causally ambiguous abstracts (see row 2, Table 2) were identical to the causally worded

**Table 1. Descriptive statistics for the demographic variables of the full survey sample.**

| Variable | | n | % | M (SD) |
|---|---|---|---|---|
| Condition | A | 36 | 25 | |
| | B | 35 | 25 | |
| | C | 34 | 24 | |
| | D | 35 | 25 | |
| Source | cogdevsoc listserv | 67 | 47 | |
| | PsychMad | 40 | 28 | |
| | Twitter | 35 | 25 | |
| Sex | Male | 43 | 30 | |
| | Female | 77 | 54 | |
| | Non-binary | 3 | 2 | |
| | Prefer not to answer / Do not answer | 19 | 3 | |
| Age | | 120 | | 32.98 (9.23) |
| Statistics Courses Taken | | 127 | | 4.13 (2.74) |
| Country Academic Career | United States | 83 | 59 | |
| | Canada | 10 | 7 | |
| | Other / Did not respond | 49 | 34 | |
| Age Learned English | Native Speaker | 83 | 58 | |
| | Before Age 6 | 13 | 9 | |
| | Between Ages 7–10 | 16 | 11 | |
| | Between Ages 11–14 | 12 | 9 | |
| | Between Ages 15–18 | 1 | 1 | |
| | Prefer not to answer / Did not respond | 17 | 12 | |
| Career Status | Faculty | 46 | 32 | |
| | Postdoctoral Researcher | 28 | 20 | |
| | PhD Student | 47 | 33 | |
| | Masters Student | 2 | 1 | |
| | Undergraduate | 1 | 1 | |
| | Other / Did not respond | 18 | 13 | |
| Academic Field | Medical & Health Sciences | 1 | 1 | |
| | Psychology | 120 | 85 | |
| | Natural Sciences | 1 | 1 | |
| | Mathematics | 1 | 1 | |
| | Other / Did not respond | 22 | 16 | |
| Psychology Subfield | Clinical | 12 | 9 | |
| | Cognitive | 33 | 23 | |
| | Neuroscience | 13 | 9 | |
| | Developmental | 66 | 47 | |
| | Quantitative | 11 | 8 | |
| | Social/Personality | 16 | 11 | |
| | Educational | 8 | 6 | |
| | Other / Did not respond | 34 | 24 | |
| Passed Manipulation Check | | 63 | 44 | |
| Failed Manipulation Check | | 79 | 55 | |
| Participants Excluded | | 87 | 61 | |

Note. n = 142. Condition: (A) is Abstract 1 Statistical language, Abstract 2 Causal language, (B) is Abstract 1 Causal language, Abstract 2 Statistical language, (C) is Abstract 2 Causal language, Abstract 1 Statistical language, and (D) is Abstract 2 Statistical language, Abstract 1 Causal language. Participants that preferred not to answer questions or did not answer were reported together to prevent any possibility of identification. Some academic fields categorized as other included 6 areas that were included in the analyses as psychological subfields.

**Table 2. Study abstracts of fictitious studies used as stimuli.**

| | Abstract 1: Scientific Reasoning | Abstract 2: Reading Achievement |
|---|---|---|
| Causal Language | Does children's early scientific reasoning ability **impact** whether they will go to college? Using data from a very large (n = 10,516), nationally representative dataset, **we estimate the causal effect of** children's kindergarten entry score on a standardized test of scientific reasoning skills on their probability of attending a four-year college or university by age 21. We regress children's college attendance on their kindergarten science score, controlling for children's household income, gender, and ethnicity. We find that [a standard deviation increase in children's scientific reasoning scores **causes** a 7-percentage point increase (p < .0001) in the probability a child would attend a four-year college or university by age 21]. We conclude that raising the quantity and quality of early science instruction is an important way to raise children's educational prospects in the 21st century. (Word count: 134) | Third grade is an important year in the development of children's language and literacy skills, as children progress from "learning to read" to "reading to learn", thus reading achievement at this time may **affect** much later academic outcomes. Using data from a large (n = 3,614) administrative dataset from a large urban U.S. school district, **we estimate the causal effect of** children's grade 3 reading skills, measured by scores on a state administered reading test, on their PSAT Verbal scores in Grade 11. We regress children's PSAT Verbal scores on their grade 3 reading scores, controlling for children's household income, gender, and whether the child is a native English speaker. We find that [a standard deviation increase in children's grade 3 reading achievement scores **causes** a .4 standard deviation increase (p < .0001) in children's grade 11 PSAT Verbal score]. We conclude that extra resources for third grade reading instruction are likely to improve children's college readiness and propose this as a useful intervention."(Word count: 164) |
| Causally Ambiguous Statistical Language | Does children's early scientific reasoning ability **predict** whether they will go to college? Using data from a very large (n = 10,516), nationally representative dataset, **we test the degree to which children's kindergarten entry score on a standardized test of scientific reasoning skills predicted** their probability of attending a four-year college or university by age 21. We regress children's college attendance on their kindergarten science score, controlling for children's household income, gender, and ethnicity. We find that [a standard deviation increase in children's scientific reasoning scores **predicted** a 7-percentage point increase (p < .0001) in the probability a child would attend a four year college or university by age 21]. We conclude that raising the quantity and quality of early science instruction is an important way to raise children's educational prospects in the 21st century. (Word count: 135) | Third grade is an important year in the development of children's language and literacy skills, as children progress from "learning to read" to "reading to learn", thus reading achievement at this time may **predict** much later academic outcomes. Using data from a large (n = 3,614) administrative dataset from a large urban U.S. school district, **we test the degree to which** children's grade 3 reading skills, measured by scores on a state administered reading test, **predict** their PSAT Verbal scores in Grade 11. We regress children's PSAT Verbal scores on their grade 3 reading scores, controlling for children's household income, gender, and whether the child is a native English speaker. We find that [a standard deviation increase in children's grade 3 reading achievement scores **predicts** a .4 standard deviation increase (p < .0001) in children's grade 11 PSAT Verbal score]. We conclude that extra resources for third grade reading instruction are likely to improve children's college readiness and propose this as a useful intervention. (Word count: 164) |

*Note*. These abstracts are randomly presented to participants. Conditions are counterbalanced so that participants can be in 1 of four conditions in which abstracts can be presented as follows: (1) abstract 1 causal language & abstract 2 causally ambiguous language (2) abstract 2 causal language & abstract 1 causally ambiguous language, (3) abstract 1 causally ambiguous language & abstract 2 causal language, and (4) abstract 2 causally ambiguous language & abstract 1 causal language. Text bolded here highlights differences between conditions but was not bolded in the original stimuli.

abstracts except that causal language was replaced by causally ambiguous language; for example, the word "causes" was replaced with "predicts." For each abstract, causal wording was replaced with causally ambiguous statistical wording in 3 instances: once when the purpose of the study was described, once when the purpose of the analysis was described, and once when describing the results of the study. The abstracts concluded with policy-relevant conclusions that relied on causal interpretations of the results. These conclusions did not differ between the causal and causally ambiguous abstracts.

## Procedure

Each participant received two of the four abstracts—the causally-worded abstract for one study (e.g., scientific reasoning study) and the causally ambiguous abstract for the other study (e.g., reading achievement study). We randomized which study was presented in its causal version and whether the causal or causally ambiguous abstract was presented first, resulting in four conditions (see Table 1 and S1 Table in S1 File). Participants read the first abstract and answered Questions 1 and 2 for that abstract, then did the same for the second abstract. Next, participants answered the manipulation check, followed by the demographic questionnaire,

which was labeled "optional". We did not predict any effects involving the demographic questionnaire variables.

After participants read each abstract, they were asked the two questions. Question 1 asked "Based on the limited information available in this abstract, how would you rate the quality of the design and analysis of the study described?". Participants answered using the following scale: 5 = *Very high quality*, 4 = *High quality*, 3 = *Moderate quality*, 2 = *Low quality*, 1 = *Very low quality*. Question 2 asked "Based on the limited information available in this abstract, how strongly does the study support the conclusion in the final sentence, quoted below:". Participants responded to question 2 using the following scale: 5 = *Very strongly*, 4 = *Strongly*, 3 = *Moderately*, 2 = *Weakly*, 1 = *Very Weakly/Not at all*. The abstract and questions were shown on the same page, allowing participants to refer to the abstract as they evaluated it.

After participants read both abstracts they were asked the following question as a manipulation check: *"How many of the two abstracts you just read employed explicitly causal language in their description of the results?"* which participants answered on a scale of 0 = *Neither*, 1 = *Only one*, and 2 = *Both*. Participants were not allowed to go back to change their answer after the manipulation check. Then, participants were given a brief demographic questionnaire about academic status, field of study, and psychology subfield of study followed by six optional questions about personal background characteristics such as age, sex, and English language learner status. At the end of the survey we asked participants to *"Please leave any feedback (for example, about how to improve the survey). We take your feedback very seriously"*. All questions are provided as they were asked verbatim in the preregistration document, Appendix B. (available via https://osf.io/2v8mg/).

## Analysis

As pre-registered, we estimated random intercept models, with responses nested within participants, using maximum likelihood estimation. We use the random intercept model to allow for the variance to be clustered at the individual level since we have repeated observations improving the precision of our estimates. In addition, this analysis allows us to model a different intercept for each individual when assessing the differential impact of causal language v. causally ambiguous language on their ratings of study quality and policy conclusion. Data and analysis code are available at https://osf.io/2v8mg/. To estimate the effect of causally ambiguous statistical language wording on the key outcome measures, we regressed responses to the questions about study quality and policy relevance on an indicator of whether the abstract was causally worded, along with indicators for the order in which the abstract was seen, and which hypothetical study was described in the abstract. Our preregistered hypotheses pertained to the effect of causal (vs. causally ambiguous statistical) language. However, for clarity, we present the impacts of the reverse-coded treatment, causally ambiguous statistical (vs. causal) language.

For each rating question—perceived quality and perceived support for policy conclusions—we estimated the effect of causally ambiguous statistical language in a series of four models. Model 1 included the full study sample and both abstracts and provides the most precise estimate of the treatment effect. Model 2 included the full study sample ($n$ = 142) but only included data from the first abstract viewed by each participant. This model was included to test whether results were sensitive to order effects. Because in the real world, readers may not usually judge the merits of two abstracts that differ in their use of causal language one after the other, the estimate based on participants' ratings of the first abstract only is plausibly the most externally valid estimate. However, it is less precise because it uses only half the available data. Model 3 included the first 100 participants regardless of whether they met the inclusion criteria

($n$ = 100) as an robustness check because our preregistration plan was to stop collecting data once we gathered 100 participants. Finally, Model 4 included only the preregistered sample that met all the inclusion criteria, we refer to this model throughout the results section as representing the best test of our hypotheses ($n$ = 55). In the supplementary materials we provide several additional robustness checks.

## Results

We report descriptive statistics of the mean and standard deviation of participants responses for the full survey in Table 3. Participant responses across the experimental conditions are balanced. Overall means, indicate that participants in the causal language condition rated both abstracts as having a lower quality study design and analysis, and lower support for policy conclusions.

### Effects of causally ambiguous statistical language on perceived quality

Following our preregistration plan we first conducted confirmatory analyses to estimate the impact of causal framing on participants' ratings of study quality. Table 4 indicates that results for the key parameter estimate, "causally ambiguous statistical language", were quite similar and highly significant across the four models. Participants reliably rated abstracts worded in causally ambiguous statistical language as of higher quality than those worded in causal language, with estimates of the difference ranging from .34 to .53 points on the 5-point Likert scale. Expressed in standardized effect sizes, these estimates range from .48 to .80 (in the

**Table 3. Descriptive statistics for survey responses from the full survey sample.**

| Variable | Q1: Design & Analysis | | Q2: Support for Conclusion | |
|---|---|---|---|---|
| | $n$ | $M$ (SD) | $n$ | $M$ (SD) |
| Abstract 1: Scientific Reasoning | | | | |
| Statistically Ambiguous Language | 65 | 3.29 (0.80) | 65 | 2.31 (1.01) |
| Causal Language | 64 | 2.73 (0.82) | 64 | 1.94 (0.85) |
| Abstract 2: Reading Achievement | | | | |
| Statistically Ambiguous Language | 66 | 3.18 (0.74) | 66 | 2.18 (0.96) |
| Causal Language | 65 | 2.86 (0.77) | 65 | 2.20 (1.11) |
| Overall | | | | |
| Statistically Ambiguous Language | 131 | 3.24 (0.77) | 131 | 2.24 (0.99) |
| Causal Language | 129 | 2.80 (0.79) | 129 | 2.07 (0.99) |

Note. $n$ = 142. Q1 asks participants about the quality of the design and analysis presented in the abstract, Q2 asks participants about the strength of support for the conclusion of the abstract. Participants excluded are still included in this descriptive analysis, and later excluded in the pre-registered analytic sample. 87 cases were excluded because participants answered first two questions in under 30 seconds (n = 12), participants were not in a field related to psychology (n = 3), participants were not doctoral students, post-doctoral researchers, or faculty (n = 6), or they failed the manipulation check (n = 79); there was overlap across the participants reasons for exclusion. Question 1:"Based on the limited information available in this abstract, how would you rate the quality of the design and analysis of the study described?" is answered on a scale from 1–5 where 5 = Very high quality, 4 = High quality, 3 = Moderate quality, 2 = Low quality, and 1 = Very low quality. Question 2: "Based on the limited information available in this abstract, how strongly does the study support the conclusion in the final sentence, quoted below: "We conclude that raising the quantity and quality of (resources for third grade reading OR early science instruction) is an important way to raise children's educational prospects in the 21st century." Is answered on a scale from 5 = Very strongly, 4 = Strongly, 3 = Moderately, 2 = Weakly, 1 = Very Weakly/Not at all.

**Table 4. Participant ratings of quality of design and study.**

| Model | 1 | | | 2 | | | 3 | | | 4 | | |
|---|---|---|---|---|---|---|---|---|---|---|---|---|
| | Estimate | SE | Est. (in SD) | Estimate | SE | Est. (in SD) | Estimate | SE | Est. (in SD) | Estimate | SE | Est. (in SD) |
| Causally Ambiguous Statistical Language | 0.44*** | 0.07 | 0.54 | 0.48*** | 0.14 | 0.59 | 0.34*** | 0.08 | 0.48 | 0.53*** | 0.11 | 0.80 |
| Abstract | -0.005 | 0.07 | | 0.01 | 0.14 | | 0.09 | 0.08 | | 0.15 | 0.11 | |
| Order | 0.21** | 0.07 | | | | | 0.17* | 0.09 | | 0.20 | 0.11 | |
| Intercept | 2.71*** | 0.13 | | 2.67*** | 0.23 | | 2.66*** | 0.15 | | 2.46*** | 0.18 | |
| Participants | 132 | | | 132 | | | 90 | | | 55 | | |
| Observations | 260 | | | | | | 177 | | | 110 | | |
| $R^2$ | 0.07 | | | 0.08 | | | 0.08 | | | 0.19 | | |
| Model Specifications | | | | | | | | | | | | |
| Both abstracts | | X | | | | | | | | | | |
| First abstract only | | | | | X | | | X | | | X | |
| Passed Manipulation check | | | | | | | | | | | X | |
| First 100 participants | | | | | | | | X | | | | |
| Passed exclusionary criteria | | | | | | | | | | | X | |

Note. *** $p < 0.001$

**$p < 0.01$

*$p < 0.05$. Although our full sample size was 142, 8 cases were missing the second abstract and were excluded from the analyses.

PsychMD = Psychological Methods Discussion Facebook group. The dependent variable is Question 1:"Based on the limited information available in this abstract, how would you rate the quality of the design and analysis of the study described? "is answered on a scale from 1–5 where 5 = Very high quality, 4 = High quality, 3 = Moderate quality, 2 = Low quality, and 1 = Very low quality. Causally Ambiguous Statistical Language is a dummy variable in which causal language used in the abstract = 0 and Causally Ambiguous Statistical Language = 1. The abstract variable is coded as 1 = abstract 1 and 2 = abstract 2 to indicate effects of abstract used on quality rating. Order indicates which abstract was seen first (1) or second (2) to measure the effects of abstract order on quality rating. Model 2 is an ordinary least squares model using only the quality response from the first abstract seen by the participants. All other models use individual random effects to estimate the effect of the independent variables on changes in quality rating within-person since all participants saw to abstracts and provided two quality ratings. SD indicates one standard deviation for the outcome variable. An effect size can be calculated by dividing regression estimates by the outcome standard deviation. R2 reflects the pseudo r-squared for fixed effect. Model 4 represents the preregistered sample, 84 cases were excluded because participants answered first two questions in under 30 seconds (n = 12), participants were not in a field related to psychology (n = 6), they failed the manipulation check (n = 64), were not enrolled in a PhD program, post-doctoral position, or a faculty position(n = 1), or they were not at least 18 years-old (n = 1).

preregistered Model 4). For the full sample, results were similar when both responses were included (Model 1) as to when only the first abstract response was included (Model 2), suggesting that the within-participants design did not elicit a contrast effect that exaggerated these estimates.

Results using all available observations (Model 1) are displayed in Fig 1. As can be seen in the figure, most participants gave a different response in the two conditions, with the most common response patterns being a 4 or a 3 when studies were described in causally ambiguous statistical language, with a rating of 2 or 3 when studies were described in causal language. The higher amount of area shaded in red indicates that the most common pattern was to rate causally worded abstracts as of lower quality.

### Effects of causally ambiguous statistical language on perceived support for policy conclusions

We also conducted confirmatory analyses to estimate the impact of causal framing on participants' ratings of strength of support for policy conclusions. Table 5 indicates that results for the key parameter estimate, again "causally ambiguous statistical language", were similar

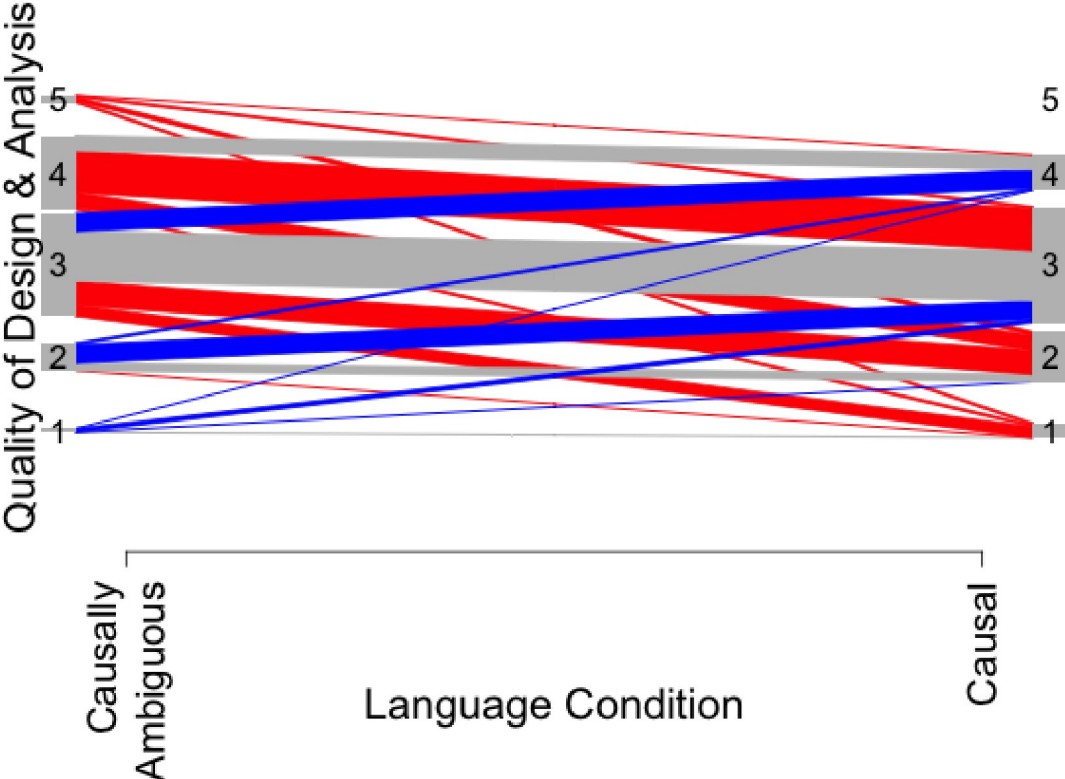

**Fig 1. Changes in ratings by condition for questions 1.** *Note. N* = 142. This panel compares ratings from question 1: "*Based on the limited information available in this abstract, how would you rate the quality of the design and analysis of the study described*? ". Lines represent differences in participant rating of quality between abstracts that have statistical language and causal language. Gray lines indicate the number of participants that rated abstracts with different languages the same. Blue lines indicate the number of participants that rated abstracts with causal language higher than abstracts with statistical language. Red lines indicate the number of participants that rated abstracts with statistical language higher than abstracts with causal language.

across the four models. Participants rated abstracts that used causally ambiguous statistical language as supporting policy conclusions as strongly, or more strongly, than causally worded abstracts in Model 1. Estimates in Models 2–4 were similar to Model 1 in magnitude but statistically nonsignificant and ranging from .16 to .25 points on the 1–5 Likert scale. The preregistered Model 4 generated the largest point estimate, but it was not statistically significant, because over half of participants were excluded. Expressed in standardized effect sizes, these estimates range from .16 to .27. The estimate was significantly greater than 0 in the full sample using all available observations (Model 1). Fig 2 shows that more participants gave lower ratings to causally worded abstracts than to statistically worded abstracts, but many participants shifted in both directions. For the full study sample, the estimate was similar whether both responses were included (Model 1) compared to when only the first abstract response was included (Model 2), suggesting that testing effects did not affect the magnitude of the estimate.

**Table 5. Participant ratings of information support for policy conclusion.**

| Model | 1 | | | 2 | | | 3 | | | 4 | | |
|---|---|---|---|---|---|---|---|---|---|---|---|---|
| | Estimate | SE | Est. (in SD) | Estimate | SE | Est. (in SD) | Estimate | SE | Est. (in SD) | Estimate | SE | Est. (in SD) |
| Causally Ambiguous Statistical Language | 0.20* | 0.08 | 0.20 | 0.20 | 0.15 | 0.23 | 0.16 | 0.10 | 0.16 | 0.25† | 0.12 | 0.27 |
| Abstract | 0.03 | 0.08 | | -0.08 | 0.15 | | 0.11 | 0.10 | | 0.05 | 0.12 | |
| Order | 0.49*** | 0.08 | | | | | 0.45*** | 0.10 | | 0.34** | 0.12 | |
| Intercept | 1.79*** | 0.15 | | 1.95*** | 0.25 | | 1.73*** | 0.18 | | 1.89*** | 0.22 | |
| Participants | 132 | | | 132 | | | 90 | | | 55 | | |
| Observations | 260 | | | | | | 177 | | | 110 | | |
| $R^2$ | 0.07 | | | 0.02 | | | 0.06 | | | 0.05 | | |
| Model Specifications | | | | | | | | | | | | |
| Both abstracts | | X | | | | | | | | | | |
| First abstract only | | | | | X | | | X | | | X | |
| Passed Manipulation check | | | | | | | | | | | X | |
| First 100 participants | | | | | | | | X | | | | |
| Passed exclusionary criteria | | | | | | | | | | | X | |

Note. *** $p < 0.001$

**$p < 0.01$

*$p < 0.05$

† $p = 0.05$. Although our full sample size was 142, 8 cases were missing the second abstract and were excluded from the analyses. Effect size is standardized to outcome standard deviations. The dependent variable is Question 2: *"Based on the limited information available in this abstract, how strongly does the study support the conclusion in the final sentence, quoted below: "We conclude that raising the quantity and quality of (resources for third grade reading* OR *early science instruction) is an important way to raise children's educational prospects in the 21st century." Is answered on a scale from 5 = Very strongly*, 4 = *Strongly*, 3 = *Moderately*, 2 = *Weakly*, 1 = *Very Weakly/Not at all*. Causal language is a dummy variable in which causal language used in the abstract = 1 and statistical language = 0. The abstract variable is coded as 1 = abstract 1 and 2 = abstract 2 to indicate effects of abstract used on support rating. Order indicates which abstract was seen first (1) or second (2) to measure the effects of abstract order on support rating. Model 2 is an ordinary least squares model using only the support response from the first abstract seen by the participants. All other models use individual random effects to estimate the effect of the independent variables on changes in support rating within-person since all participants saw to abstracts and provided two quality ratings. SD indicates one standard deviation for the outcome variable, by multiplying the estimate by the SD you can calculate the average change in rating in standard deviation units.

## Exploratory subgroup analyses

Because we used a convenience sample, external validity was a major concern. We thus estimated the effects of causally ambiguous statistical language for two kinds of subgroups: 1) the source from which a participant accessed the survey, and 2) participants' occupational status (S3-S7 Tables in S1 File). Estimated effects of causally ambiguous statistical language on ratings of study quality went in the same direction across source subgroups, although they were larger in the PsychMD and cogdevsoc subgroups than in the Twitter subgroup (S3 Table in S1 File). The effect of causally ambiguous statistical language on ratings of support for policy conclusions was only statistically significant in the cogdevsoc subgroup and was close to 0 in the other groups (S4 Table in S1 File). The cogdevsoc subgroup represents researchers in various psychology subfields including developmental, educational, cognitive, and social/personality psychology subfields (S5 Table in S1 File). Estimates for both questions were similar for psychology faculty, postdocs, and Ph.D. students (S6 and S7 Tables in S1 File).

## Exploratory participant self-reports

To probe participants' thinking, we read through participants' feedback provided at the end of the survey. Although we did not ask participants to explain the reasoning behind their ratings,

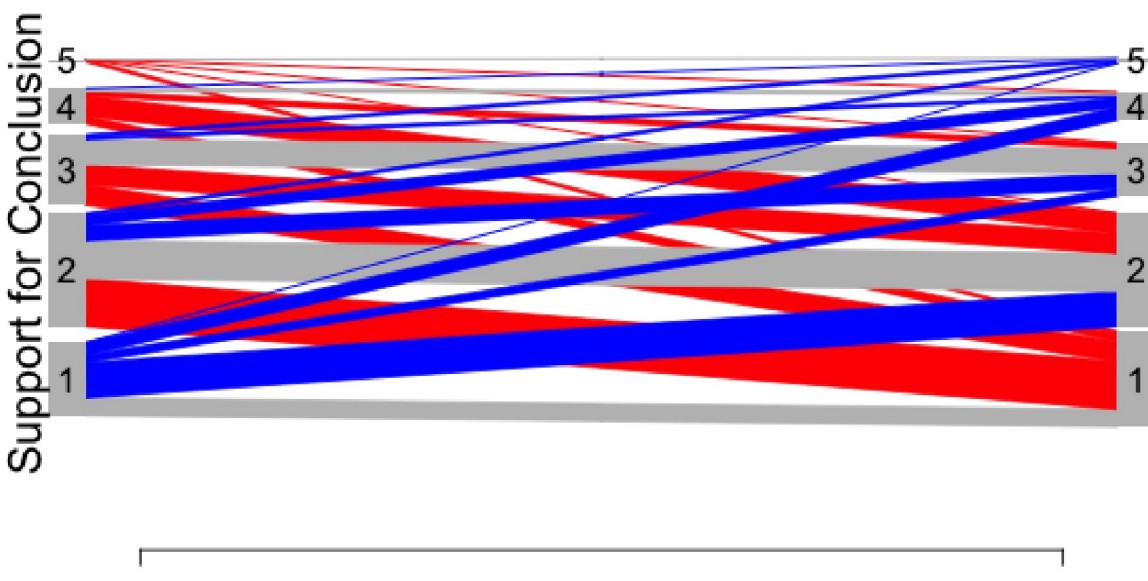

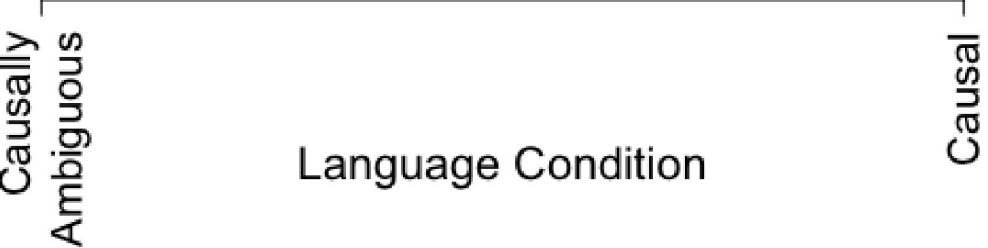

**Fig 2. Changes in ratings by condition for question 2.** *Note. N* = 142. This panel compares ratings from question 2: *"Based on the limited information available in this abstract, how strongly does the study support the conclusion in the final sentence, quoted below*: *"We conclude that raising the quantity and quality of (resources for third grade reading* OR *early science instruction) is an important way to raise children's educational prospects in the 21st century."*. Lines represent differences in participant rating of quality between abstracts that have statistical language and causal language. Gray lines indicate the number of participants that rated abstracts with different languages the same. Blue lines indicate the number of participants that rated abstracts with causal language higher than abstracts with statistical language. Red lines indicate the number of participants that rated abstracts with statistical language higher than abstracts with causal language.

some participants self-reported their reasoning in an open-ended request for feedback at the end of the survey. Of the 142 participants, 22 provided feedback on the survey. Of these, 10 were substantive critiques of the contents of the abstracts, and the remaining comments provided helpful critiques of the structure of the survey or compliments.

We conducted an exploratory analysis of participant feedback to probe their reasoning about the quality of abstract design and support for policy implications. For example, of the 10 critiques of abstract content, three participants thought that the conclusion for both abstracts claimed causality and two of them stated these claims were wrong; two other participants discussed confounding variables, and one participant believed that the quality of both studies was

inadequate to answer a causal question but adequate to answer a predictive question. Consistent with the idea that causally ambiguous statistical language can communicate causal information, three critiques expressed concerns about the causal conclusions of both abstracts. One participant was concerned about the norm against causal language being violated and therefore rated the abstract about reading achievement written with causal language as lower quality. However, the same participant found that the other abstract about scientific reasoning written in causally ambiguous language made policy conclusions that were higher quality due to a larger sample size; thus, the norm against causal language did not appear to preclude them from endorsing the causal inferences made by the policy conclusions. Lastly, other factors such grammar and run-on sentences, informed one participant's ratings of study design quality. This exploratory analysis offers converging evidence that several participants based their decisions in part on perceptions of the extent to which the abstract explicitly described findings as causally informative.

## Discussion

The current paper demonstrates that participants more favorably rated the study design quality of abstracts that followed the norm against the use of causal language in psychological research without reducing their perceived support for conclusions that implicitly rely on causal interpretation of results. This evidence highlights the need to rethink how causal language can be used to meet the goals of nonexperimental psychologists to test theories that include causal hypotheses. As hypothesized, participants rated studies described with causally ambiguous statistical language as of higher quality (by .34-.80 effect size standardized to the outcome variable), and as similarly supportive or more supportive (by .16-.27 effect size standardized to the outcome variable) of policy recommendations. Thus, under some circumstances, causally ambiguous statistical language may allow authors to communicate causal interpretations without being punished for violating the norm against straightforward causal language.

Effects on perceived support for policy recommendations were smaller and less robust than effects on perceived study quality. However, to reach the eyes of the public and policymakers, a study must be published, and publication likely depends on evaluations of study quality. By increasing the chance of publication, causally ambiguous language could increase the chance of policy recommendations being widely disseminated.

### Why are authors making policy recommendations that are not supported by the ambiguous causal language of the results?

Recommendations for policy and practice that are based on nonexperimental research are common in educational psychology [16]. Concerned researchers have advocated for policies that restrict discussions of practical and policy implications that extend beyond the evidence from the data [17–20]. For example, the authors warn against making recommendations that generalize study results to populations that were not included in the sample [21–23]. We agree with the authors that policy recommendations should be appropriately limited in scope.

However, as noted in the introduction section, nonexperimental psychology contains frequent examples, most obviously studies attempting to estimate the effects of programs or policies in the absence of random assignment, for which reporting associations without discussing policy or causation makes little sense. This is not an argument for naïvely assuming that all our estimates have a clearly understood causal interpretation: Rather, we maintain that for researchers to test theory and provide policy-relevant empirical evidence, authors of nonexperimental studies should explicitly state their intention to test hypotheses about causal mechanisms, attempt to rule out plausible alternative hypotheses, and calibrate the strength of

the causal evidence to the strength of the policy conclusion. Merely replacing causal language with causally ambiguous statistical language is no substitute for these difficult but essential steps. Researchers should consider using causal language, when the assumptions of an analysis are explicit, and sources of bias recognized. This is much more useful scientific endeavors than sweeping those details under the veil of statistically ambiguous language and proceeding boldly with the inference/policy recommendations. We thank an anonymous reviewer for making this point.. We encourage future research on how causal language can best be employed to communicate researchers' communication about psychological research in ways that address the underlying theory under consideration but do not convey overconfidence about causal implications.

### Why does causally ambiguous statistical language increase ratings of study quality and support for policy conclusions?

The reasons why participants rated abstracts with similar research designs and conclusions as having lower quality and being less supportive of policy implications when they were written with causal language as opposed to causally ambiguous statistical language are not clear. Possible explanations include that readers view the use of causal language to describe results of non-experimental studies as unfamiliar or as a norm violation. Several participants responded to an open-ended prompt for feedback on the study with reference to the norm, suggesting this was an important factor contributing to the observed effects. Another possible reason is that causal language makes plausible alternative explanations more salient to the reader. Indeed, some participants noted that failing to control for additional confounds made the hypothetical studies inadequate to answer a causal question. Finally, the negative effect on perceived support for policy recommendations may reflect a halo effect, wherein any of the above factors make readers view a nonexperimental study described with causal language as less trustworthy and less useful in a variety of ways.

Regardless of the mechanisms, it is concerning that participants rated an abstract with the same research design and conclusion as being as or more supportive of policy recommendations when it was worded in accordance with the norm against causal language. Whether the support for policy recommendations is viewed differently due to the salience of methodological limitations or a halo effect, the conclusion drawn is the same, readers favored a non-experimental design more when the language used was causally ambiguous. This is an important consideration for discussions around optimal norms pertaining to the use of causal language in nonexperimental psychology.

### Implications: Should we encourage causal language?

Perhaps a useful practice for improving causally informative statistical analysis in psychology would be to encourage psychologists testing causal theories to use causal language throughout their manuscripts. Pearl and Mackenzie asserted that statisticians' reluctance to incorporate causal language into their work limited the types of questions they could answer and resulted in longstanding apparent paradoxes [11]. Hernán describes the problem of specifying a model without a clear causal question in mind using the example of estimating the association between drinking wine and heart disease: If the researcher's ultimate goal were to test the statistical association between these variables, then measuring and statistically adjusting for likely confounders would not be necessary [9, 10]. However, as Rutter [24] argued in response to the argument that nonexperimental studies provide important information about associations that have no causal meaning, "it is difficult to see why anyone would be interested in statistical associations or correlations if the findings were not in some way relevant to an understanding

of causative mechanisms" (p. 377). Therefore, although retreating to the motte (the claim that the researcher is only interested in associations) may protect them from criticism in the short-term, the ultimate goal of understanding how wine consumption influences heart disease would be better served by a straightforward discussion of the assumptions required to address the causal question of interest: What is the impact of wine on heart disease? For this reason, several have argued that psychologists' reluctance to explicitly talk about causality limits the questions they can address directly, and results in seemingly contradictory findings, stifling cumulative progress in psychology [1, 13, 14, 24, 25].

The possible benefits of encouraging psychologists to specify their causal theories explicitly include clarity to readers and clarity to writers. Causal language can make the link between research methods and theory more transparent to support the responsible use of causal methods to test the causal theories. Most findings in nonexperimental psychology are consistent with multiple possible theories[8, 26, 27], often with different policy implications. Encouraging researchers testing causal theories to use causal language throughout the manuscript can help clarify the assumptions, which, along with these findings, led the writer to a particular set of theoretical and policy conclusions.

This approach is used in other fields where results from studies with strong quasi-experimental designs are often described with straightforward causal language as well [28]. Further, use of causal language could make both writers and readers more alert to potential biases in estimates of causal effects (e.g., over-estimation of effect size due to the presence of unobserved confounding variables). Careful thinking about these details is not a fool's errand: Researchers have identified several instances in which, through careful design and analysis, nonexperimental analyses have yielded empirical estimates very similar to impacts estimated from randomized experiments [21, 29].

Although we find these arguments compelling, it does not necessarily follow that causal language will improve standards or practices around the presentation of causal evidence in psychology. For the benefits of causal language to outweigh the costs, using such language must cause researchers to *think* differently. Without stronger norms around causal inference (e.g., different training, longer manuscripts with more robustness checks and falsification tests), dissolving the norm against straightforward causal language in psychology could backfire, resulting in stronger claims *without* stronger evidence. Journals might consider incentives (e.g., badges of recognition) for researchers using rigorous methods for causal inference. Also, to counter "hedge-drift"—wherein key study limitations are relegated to the middle of long discussion sections, after the strongest claims have already been made—journals might require important limitations and caveats to any major claims in the paper to appear in titles and abstracts [22, 30, 31].

## Limitations

The generalizability of these results is limited in three ways. First, our convenience sample is not representative of psychological researchers in general or in any subfield. While subgroup analyses suggested that participants from different recruiting sources, in different research stages and different psychological subfields, provided similar ratings of study quality, it is possible that certain subfields have different norms. The exploratory subgroup analyses suggested that support for the conclusion in the final sentence of the abstract was only significantly affected within participants recruited from one of the three sources (see S4 Table in S1 File).

Second, we only tested two different types of abstracts with the term "predict" reflecting the causally ambiguous condition. A potential issue with using the term "predict" is that individuals may have interpreted it as reflecting a study exclusively concerned with predictive

modelling. This likely explains the high error rate on our manipulation check. Future work might benefit from using words such as "associate", and "link" which Haber and colleagues [17] identified as being more common and causally ambiguous than the term "predict"– although at least half of reviewers rated all three of these words as suggesting a causal link. Whether this would improve nonexperimental research on net would depend on the extent to which these words increase perceived study quality yet continue to convey some degree of causal information to the reader. Finally, perhaps most important for guiding best practices related to causal language is the extent to which wording affects the author's thinking about causal assumptions and interpretations, which we hope future work will measure.

Third, perhaps with stronger study designs, a different pattern of finding would emerge. On average, participants found the abstracts to somewhat weakly support the policy conclusions across the two conditions (mean rating = 2.16 on a 5-point scale). Creating an important limitation of this experiment in that it is difficult to make a judgment about whether participants were overly harsh in their ratings of abstracts described with straightforward causal language, overly generous in their ratings of abstracts described with causally ambiguous statistical language, or both. This is a difficult question that should bear strongly on the implications of these findings.

One potential objection to our conclusions is that it was not the causal language, but rather the inability of the methods described in these hypothetical abstracts to support causal claims, which raters penalized in their ratings of study quality and support for policy conclusions in the causal language condition. However, we do not view these as alternative explanations: The use of causal language was penalized *because* it makes clear that the method does not provide strong support for the conclusions. To be clear, we do not hypothesize that our findings would generalize to studies with stronger designs for causal inference.

## Conclusion

Participants reliably rated the quality of the study design and analysis as higher when the abstract was worded in causally ambiguous statistical language than when it was worded in straightforward causal language. Furthermore, participants reported that policy recommendations were as well supported or better supported when causally ambiguous statistical language was used, although such recommendations are only justified in the presence of a causal effect. Taken together, we find that using causally ambiguous language instead of straightforward causal language did not prevent participants from viewing nonexperimental work as supporting policy conclusions. Future studies should be conducted to examine mechanisms responsible for this effect. In the meantime, we strongly discourage researchers from merely scrolling through their manuscripts and replacing instances of the word "predicts" with "causes". Rather, we hope psychologists will consider causal thinking from the start, perhaps adopting Hernán's recommendations: to articulate clear causal questions (using straightforward causal language), distinguish them from the procedure used to emulate the causal question in the study, and pinpoint or bound estimates by triangulating results from different studies [10].

## Supporting information

**S1 File.**
(DOCX)

## Acknowledgments

The authors thank the participants for volunteering their time to complete the study. We thank Mayan Castro for her help with survey design and piloting. We thank Julia Rohrer for helpful comments on a previous version and Michael Grosz for a helpful review. We thank Ulrich Schimmack for help with posting to social media. We thank Lisa Fazio, Nora Newcombe, Daniel Ansari, Ruben Arslan, Sarah Pressman, and Kevin King for posting about this study on social media. We thank Greg Duncan, Jade Jenkins, Tyler Watts and Paul Hanselman for helpful discussions about this project.

## Author Contributions

**Conceptualization:** Daniela Alvarez-Vargas, David Braithwaite, Hugues Lortie-Forgues, Melody Moore, Sirui Wan, Drew Hal Bailey.

**Data curation:** Daniela Alvarez-Vargas, David Braithwaite, Hugues Lortie-Forgues, Drew Hal Bailey.

**Formal analysis:** Daniela Alvarez-Vargas.

**Investigation:** Daniela Alvarez-Vargas.

**Methodology:** Daniela Alvarez-Vargas, David Braithwaite, Hugues Lortie-Forgues, Sirui Wan, Elizabeth Martin, Drew Hal Bailey.

**Project administration:** Daniela Alvarez-Vargas, David Braithwaite, Drew Hal Bailey.

**Supervision:** David Braithwaite, Hugues Lortie-Forgues, Drew Hal Bailey.

**Validation:** Daniela Alvarez-Vargas, David Braithwaite, Hugues Lortie-Forgues, Drew Hal Bailey.

**Visualization:** Daniela Alvarez-Vargas, Drew Hal Bailey.

**Writing – original draft:** Daniela Alvarez-Vargas, Drew Hal Bailey.

**Writing – review & editing:** Daniela Alvarez-Vargas, David Braithwaite, Hugues Lortie-Forgues, Melody Moore, Sirui Wan, Elizabeth Martin, Drew Hal Bailey.

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
