## [Decision Letter · Decision Letter 0]

22 Nov 2022

PONE-D-22-24784Hedges, mottes, and baileys: Causally ambiguous statistical language can increase perceived study quality and policy relevancePLOS ONE

Dear Dr. Alvarez-Vargas,

Thank you for submitting your manuscript to PLOS ONE. After careful consideration, we feel that it has merit but does not fully meet PLOS ONE’s publication criteria as it currently stands. Therefore, we invite you to submit a revised version of the manuscript that addresses the points raised during the review process.

Please note that we have only been able to secure a single reviewer to assess your manuscript. We are issuing a decision on your manuscript at this point to prevent further delays in the evaluation of your manuscript. Please be aware that the editor who handles your revised manuscript might find it necessary to invite additional reviewers to assess this work once the revised manuscript is submitted. However, we will aim to proceed on the basis of this single review if possible. The reviewer has identified several important opportunities to improve the manuscript, and we would ask you to respond carefully to each of the points they have raised when preparing your revisions.

We look forward to receiving your revised manuscript.

Kind regards,

Jamie Males

Editorial Office

PLOS ONE

Journal Requirements:

2. You indicated that ethical approval was not necessary for your study. Could you please provide confirmation from your institutional review board or research ethics committee (e.g., in the form of a letter or email correspondence) that ethics review was not necessary for this study? Please include a copy of the correspondence as an ""Other"" file.

Reviewers' comments:

Reviewer's Responses to Questions

**Comments to the Author**

1. Is the manuscript technically sound, and do the data support the conclusions?

Reviewer #1: Partly

2. Has the statistical analysis been performed appropriately and rigorously? 

Reviewer #1: Yes

3. Have the authors made all data underlying the findings in their manuscript fully available?

Reviewer #1: Yes

4. Is the manuscript presented in an intelligible fashion and written in standard English?

Reviewer #1: Yes

5. Review Comments to the Author

Reviewer #1: The manuscript investigates how abstracts that describe their findings with causally ambiguous language (versus causal explicit language) are perceived and evaluated by expert readers (psychology faculty, postdocs, and doctoral students). I think this is an interesting and important study because it informs researchers about the consequences of using causally ambiguous language in reports about nonexperimental research. I also liked that they used random intercepts in their data analysis to deal with the fact that responses were nested within participants. I have reviewed a previous version of the manuscript for another journal. I was happy to see that the authors heeded several of the points that I made in my last review. Here are some comments that might help to further improve the manuscript:

(1) I agree with almost everything stated in the Introduction. However, my impression was that the Introduction section was a bit too long, repetitive, and rambling. I feel like the Introduction could and should be streamlined. For example, I would omit information from the Introduction that is not essential for the current research.

(2) The following paragraph—which consists of only one sentence—was not very clear to me (especially the part about “focusing more on their preferred causal interpretation than on ruling out plausible alternatives”): “Other common ways in which psychologists may implicitly communicate a preferred causal interpretation while hedging to avoid criticism for making causal claims on the basis of correlational data include 1) focusing more on their preferred causal interpretation than on ruling out plausible alternatives, or 2) designing studies for which results are far more interesting or important if they are causally informative.“ I would remove or revise this paragraph.

(3) Method section (page 11): “encompassing a variety of interdisciplinary subfields, as shown in Table 2.” I think the authors meant Table 1 (not Table 2).

(4) Table 2: The two versions of each abstract differ in terms of causal language at three places. Only two of the three differences are bolded in Table 2. I think the third difference (“impact” vs. “predict” and “affect” vs. “predict”, respectively) should be bolded as well.

(5) The page number is missing at some pages after page 14.

(6) Page 23: I think something is wrong with the following sentence: “Participants rated abstracts that used causally ambiguous statistical language as supporting policy conclusions as strongly as more strongly than causally worded abstracts in Model 1 […] .“ Is there a typo in the sentence? Is there an “or” missing between “as strongly” and “as more strongly”? The sentence is also very long—I didn’t quote the entire sentence. I would recommend to revise the original sentence and make two or three sentences out of it. In general, I would try to avoid sentences that go over more than 2-3 lines.

(7) Page 33: Did the authors actually mean “ratings of policy relevance” in the following sentence? “indeed, exploratory subgroup analyses suggested that ratings of policy relevance were only affected within participants recruited from one of the three sources“

I guess what the authors actually wanted to say is “support for the conclusion in the final sentence.”

(8) Page 34: The following sentence was too unspecific: “Participants reliably rated abstracts worded in causally ambiguous statistical language as of higher quality than causally worded abstract” Participants did not rate the quality of the abstracts, did they? They rated the quality of the study design and analysis.

(9) Similarly, the following text in the Abstract was too unspecific: “[participants] rated abstracts from hypothetical studies with causally ambiguous statistical language as of higher quality“ They rated the quality of the study design and analysis (not the quality of the abstract).

(10) Similarly, I found the second sentence of the Abstract in need of revision. Here is the original version of the sentence: “However, this norm may lead to higher ratings of study quality and greater acceptance of policy recommendations that rely on causal interpretations of nonexperimental findings.“ I would be clearer and more accurate to write: “However, causal ambiguous language may inhibit a critical evaluation of the study design and analysis and lead to a greater acceptance of policy recommendations that rely on causal interpretations of nonexperimental findings.“

(11) I found the following sentence on page 35 difficult to understand: “Taken together, the current study found that the causally ambiguous language did not prevent participants from viewing nonexperimental work as supporting policy conclusions that rely on causal evidence, to the same extent as the same work when described with straightforward causal language.” I think it would be good to revise this sentence.

(12) I noticed that the authors quote the preprint of the taboo paper in the References section. The manuscript has been published at a journal in the meantime: see https://doi.org/10.1177/1745691620921521

Hope that helps,

Michael P. Grosz

6. PLOS authors have the option to publish the peer review history of their article (what does this mean?). If published, this will include your full peer review and any attached files.

Reviewer #1: **Yes: **Michael P. Grosz

---

## [Author Response · Author response to Decision Letter 0]

13 Dec 2022

PONE-D-22-24784

Hedges, mottes, and baileys: Causally ambiguous statistical language can increase perceived study quality and policy relevance

PLOS ONE

Dear Dr. Jamie Males,

Thank you for giving us the opportunity to submit a revised version of our manuscript PONE-D-22-24784 “Hedges, mottes, and baileys: Causally ambiguous statistical language can increase perceived study quality and policy relevance.” We sincerely appreciate the detailed and helpful feedback you provided. 

Below, we describe how we addressed the reviewer’s comments. For the reviewer’s convenience, we have included all the reviewers’ comments, with our replies and relevant quotes from the revised manuscript following in italics. 

Sincerely,

Daniela Alvarez-Vargas

 

 Review Comments to the Author

Reviewer #1: The manuscript investigates how abstracts that describe their findings with causally ambiguous language (versus causal explicit language) are perceived and evaluated by expert readers (psychology faculty, postdocs, and doctoral students). I think this is an interesting and important study because it informs researchers about the consequences of using causally ambiguous language in reports about nonexperimental research. I also liked that they used random intercepts in their data analysis to deal with the fact that responses were nested within participants. I have reviewed a previous version of the manuscript for another journal. I was happy to see that the authors heeded several of the points that I made in my last review. Here are some comments that might help to further improve the manuscript:

(1) I agree with almost everything stated in the Introduction. However, my impression was that the Introduction section was a bit too long, repetitive, and rambling. I feel like the Introduction could and should be streamlined. For example, I would omit information from the Introduction that is not essential for the current research.

 We have omitted information from the introduction section that was not essential and now our introduction extends from page 4 to page 9. 

(2) The following paragraph—which consists of only one sentence—was not very clear to me (especially the part about “focusing more on their preferred causal interpretation than on ruling out plausible alternatives”): “Other common ways in which psychologists may implicitly communicate a preferred causal interpretation while hedging to avoid criticism for making causal claims on the basis of correlational data include 1) focusing more on their preferred causal interpretation than on ruling out plausible alternatives, or 2) designing studies for which results are far more interesting or important if they are causally informative.“ I would remove or revise this paragraph.

This paragraph has been revised and now reads as follows: In summary, common ways in which psychologists may communicate a preferred causal interpretation while hedging against causal claims is by 1) focusing more on their preferred causal interpretation than on ruling out plausible alternatives, or 2) designing studies for which results are far more interesting or important if they are causally informative. (p. 8)

(3) Method section (page 11): “encompassing a variety of interdisciplinary subfields, as shown in Table 2.” I think the authors meant Table 1 (not Table 2).

We have made this correction on page 11. 

(4) Table 2: The two versions of each abstract differ in terms of causal language at three places. Only two of the three differences are bolded in Table 2. I think the third difference (“impact” vs. “predict” and “affect” vs. “predict”, respectively) should be bolded as well.

We have boldened the words, accordingly, thank you for your attention to this! 

(5) The page number is missing at some pages after page 14.

We have corrected this mistake.

(6) Page 23: I think something is wrong with the following sentence: “Participants rated abstracts that used causally ambiguous statistical language as supporting policy conclusions as strongly as more strongly than causally worded abstracts in Model 1 […] .“ Is there a typo in the sentence? Is there an “or” missing between “as strongly” and “as more strongly”? The sentence is also very long—I didn’t quote the entire sentence. I would recommend to revise the original sentence and make two or three sentences out of it. In general, I would try to avoid sentences that go over more than 2-3 lines.

We have revised the original sentence and the new one on page 22 is as follows: Participants rated abstracts that used causally ambiguous statistical language as supporting policy conclusions as strongly, or more strongly, than causally worded abstracts in Model 1. Estimates in Models 2-4 were similar to Model 1 in magnitude but statistically nonsignificant and ranging from .16 to .25 points on the 1-5 Likert scale. The preregistered Model 4 generated the largest point estimate, but it was not statistically significant, because over half of participants were excluded. 

(7) Page 33: Did the authors actually mean “ratings of policy relevance” in the following sentence? “indeed, exploratory subgroup analyses suggested that ratings of policy relevance were only affected within participants recruited from one of the three sources“

I guess what the authors actually wanted to say is “support for the conclusion in the final sentence.”

We have made this correction at the bottom of page 32: “The exploratory subgroup analyses suggested that support for the conclusion in the final sentence of the abstract was only significantly affected within participants recruited from one of the three sources (see Table S4).”

(8) Page 34: The following sentence was too unspecific: “Participants reliably rated abstracts worded in causally ambiguous statistical language as of higher quality than causally worded abstract” Participants did not rate the quality of the abstracts, did they? They rated the quality of the study design and analysis.

This is correct; thank you for pointing this out. The sentence on at the bottom of page 33 now reads: “Participants reliably rated the quality of the study design and analysis as higher when the abstract was worded in causally ambiguous statistical language than when it was worded in straightforward causal language.”

(9) Similarly, the following text in the Abstract was too unspecific: “[participants] rated abstracts from hypothetical studies with causally ambiguous statistical language as of higher quality“ They rated the quality of the study design and analysis (not the quality of the abstract).

We made this change in our abstract.

(10) Similarly, I found the second sentence of the Abstract in need of revision. Here is the original version of the sentence: “However, this norm may lead to higher ratings of study quality and greater acceptance of policy recommendations that rely on causal interpretations of nonexperimental findings.“ I would be clearer and more accurate to write: “However, causal ambiguous language may inhibit a critical evaluation of the study design and analysis and lead to a greater acceptance of policy recommendations that rely on causal interpretations of nonexperimental findings.

We have made the sentence change as suggested. 

(11) I found the following sentence on page 35 difficult to understand: “Taken together, the current study found that the causally ambiguous language did not prevent participants from viewing nonexperimental work as supporting policy conclusions that rely on causal evidence, to the same extent as the same work when described with straightforward causal language.” I think it would be good to revise this sentence.

We have revised this sentence on page 33 and the sentence now reads as follows: Taken together, we find that using causally ambiguous language instead of straightforward causal language did not prevent participants from viewing nonexperimental work as supporting policy conclusions that rely on causal evidence . 

(12) I noticed that the authors quote the preprint of the taboo paper in the References section. The manuscript has been published at a journal in the meantime: see https://doi.org/10.1177/1745691620921521

Thank you for this citation we have updated our references.

---

## [Decision Letter · Decision Letter 1]

17 Jan 2023

PONE-D-22-24784R1Hedges, mottes, and baileys: Causally ambiguous statistical language can increase perceived study quality and policy relevancePLOS ONE

Dear Dr. Alvarez-Vargas,

Thank you for submitting your manuscript to PLOS ONE. After careful consideration, we feel that it has merit but does not fully meet PLOS ONE’s publication criteria as it currently stands. Therefore, we invite you to submit a revised version of the manuscript that addresses the points raised during the review process.

We look forward to receiving your revised manuscript.

Kind regards,

Diego A. Forero, MD; PhD

Academic Editor

PLOS ONE

Journal Requirements:

Reviewers' comments:

Reviewer's Responses to Questions

**Comments to the Author**

1. If the authors have adequately addressed your comments raised in a previous round of review and you feel that this manuscript is now acceptable for publication, you may indicate that here to bypass the “Comments to the Author” section, enter your conflict of interest statement in the “Confidential to Editor” section, and submit your "Accept" recommendation.

Reviewer #1: (No Response)

Reviewer #2: (No Response)

2. Is the manuscript technically sound, and do the data support the conclusions?

Reviewer #1: (No Response)

Reviewer #2: Partly

3. Has the statistical analysis been performed appropriately and rigorously? 

Reviewer #1: (No Response)

Reviewer #2: Yes

4. Have the authors made all data underlying the findings in their manuscript fully available?

Reviewer #1: (No Response)

Reviewer #2: Yes

5. Is the manuscript presented in an intelligible fashion and written in standard English?

Reviewer #1: (No Response)

Reviewer #2: Yes

6. Review Comments to the Author

Reviewer #1: I would like to thank the authors for addressing the points that I raised. I was satisfied with how they addressed all my points except for how they addressed point #2.

Here is the original point #2:

“(2) The following paragraph—which consists of only one sentence—was not very clear to me (especially the part about “focusing more on their preferred causal interpretation than on ruling out plausible alternatives”): “Other common ways in which psychologists may implicitly communicate a preferred causal interpretation while hedging to avoid criticism for making causal claims on the basis of correlational data include 1) focusing more on their preferred causal interpretation than on ruling out plausible alternatives, or 2) designing studies for which results are far more interesting or important if they are causally informative.“ I would remove or revise this paragraph.”

Here is how the authors responded:

“This paragraph has been revised and now reads as follows: In summary, common ways in which psychologists may communicate a preferred causal interpretation while hedging against causal claims is by 1) focusing more on their preferred causal interpretation than on ruling out plausible alternatives, or 2) designing studies for which results are far more interesting or important if they are causally informative. (p. 8)”

I was surprised that the authors wrote “in summary” at the beginning of the revised sentence because the sentence does, in my opinion, not or only partially summarize what the authors stated earlier in the text of the paragraph/Introduction. I do not think that the authors talk about “focusing more on their preferred causal interpretation than on ruling out plausible alternatives” earlier in the paragraph/Introduction. Earlier in the paragraph, the authors talked about ambiguous causal language that allows researchers to entertain an interesting causal interpretation of a finding while it allows them to retreat to a boring but more easily defendable noncausal interpretation. Did the authors also talk about “designing studies for which results are far more interesting or important if they are causally informative”? Relatedly, it was not completely clear to me (a) what these two “common ways” (1 & 2) refer to exactly. If these two ways are indeed distinct from the ways described earlier in the manuscript, I think it would be good to more clearly describe each of the two ways and perhaps provide an example for each way. Perhaps it might then be good to devote an entire paragraph to these two additional common ways in which psychologists may communicate a preferred causal interpretation while hedging against causal claims. If the two ways are not distinct from what was said earlier in the manuscript, then I would suggest revising the description of the two ways so that it becomes clear that these two ways are similar to what was said earlier. A third alternative option would be to delete the entire sentence from the manuscript. I think the Introduction would also work without this sentence.

hope that helps,

Michael P. Grosz

Reviewer #2: This manuscript provides an interesting consideration of causal wording and the implications for a readers’ interpretation of the study quality. The study is well designed and the analyses are appropriate. My main concerns are the proportion of participants who did not pass the manipulation question and the use of “predict” as the non-causal word choice in the sample abstracts.

I found the discussion of predictive models in the introduction section a bit inconsistent and therefore confusing. As the authors’ note in the introduction (no line numbers were included in manuscript), if the purpose of the model is explicitly predictive, then the objective is not to imply an underlying causal mechanism. With this, I agree. However, in several places in the introduction, it is implied that predictive models should be included in a discussion causal wording, presumably because those producing predictive models might use them for intervention purposes? Perhaps this goes back to a need for authors of non-experimental studies to be explicit as to the objective of their study. If it is to predict an outcome or to (just) identify associations for further study, then causation is not an aim and it is not necessary to control for confounding (although, the misuse of causal language may still be an issue for these studies). Based on this, I think it is unfortunate that the authors’ chose to use “predict” as their non-causal word in the abstracts evaluated by participants. Is it possible that some of the differences in perceived quality are based on the reader’s understanding of the study purpose (predictive versus causal inference), rather that the causal versus non-causal wording?

Sentence at end of page 4 starting with “Overall, impeding progress ….” Is an incomplete sentence. In the next sentence, “though” should be “although”.

To me, one of the more interesting findings of this manuscript was the proportion of individuals who erred on the manipulation question was approximately half. This was only superficially discussed but could relate to a general lack of understanding of what words have causal implications. It also may relate to my previous comment on the choice to use “predicts” as the non-causal wording in the abstracts, rather than “associated with” or some other phrase that isn’t associated with a different type of study objective (like predictive modelling). The authors did consider the implications by conducting different model specifications using (as one model) only participants passing this question. Nonetheless, the issue of word choice and the question of why so many participants did not pass the manipulation question could be more comprehensively discussed.

7. PLOS authors have the option to publish the peer review history of their article (what does this mean?). If published, this will include your full peer review and any attached files.

Reviewer #1: **Yes: **Michael P. Grosz

Reviewer #2: No

---

## [Author Response · Author response to Decision Letter 1]

2 Feb 2023

Reviewer #1: I would like to thank the authors for addressing the points that I raised. I was satisfied with how they addressed all my points except for how they addressed point #2.

Here is the original point #2:

“(2) The following paragraph—which consists of only one sentence—was not very clear to me (especially the part about “focusing more on their preferred causal interpretation than on ruling out plausible alternatives”): “Other common ways in which psychologists may implicitly communicate a preferred causal interpretation while hedging to avoid criticism for making causal claims on the basis of correlational data include 1) focusing more on their preferred causal interpretation than on ruling out plausible alternatives, or 2) designing studies for which results are far more interesting or important if they are causally informative.“ I would remove or revise this paragraph.”

Here is how the authors responded:

“This paragraph has been revised and now reads as follows: In summary, common ways in which psychologists may communicate a preferred causal interpretation while hedging against causal claims is by 1) focusing more on their preferred causal interpretation than on ruling out plausible alternatives, or 2) designing studies for which results are far more interesting or important if they are causally informative. (p. 8)”

I was surprised that the authors wrote “in summary” at the beginning of the revised sentence because the sentence does, in my opinion, not or only partially summarize what the authors stated earlier in the text of the paragraph/Introduction. I do not think that the authors talk about “focusing more on their preferred causal interpretation than on ruling out plausible alternatives” earlier in the paragraph/Introduction. Earlier in the paragraph, the authors talked about ambiguous causal language that allows researchers to entertain an interesting causal interpretation of a finding while it allows them to retreat to a boring but more easily defendable noncausal interpretation.

 Did the authors also talk about “designing studies for which results are far more interesting or important if they are causally informative”? Relatedly, it was not completely clear to me (a) what these two “common ways” (1 & 2) refer to exactly. If these two ways are indeed distinct from the ways described earlier in the manuscript, I think it would be good to more clearly describe each of the two ways and perhaps provide an example for each way. 

Perhaps it might then be good to devote an entire paragraph to these two additional common ways in which psychologists may communicate a preferred causal interpretation while hedging against causal claims. If the two ways are not distinct from what was said earlier in the manuscript, then I would suggest revising the description of the two ways so that it becomes clear that these two ways are similar to what was said earlier. A third alternative option would be to delete the entire sentence from the manuscript. I think the Introduction would also work without this sentence.

hope that helps,

Michael P. Grosz

The reviewer’s suggestions here were very helpful. We have edited the sentence and also clarified how the two common ways psychologists communicate causal interpretations while hedging against causal claims were explained in earlier paragraphs. Please find the modified paragraphs shown below with examples showcasing how psychologists use these strategies. 

“By contrast, a researcher who claims they are interested in which kinds of early skills “predict” later achievement may design a study based on its causal interpretation while only analyzing data in such a way that supports the causal interpretation without ruling out plausible alternatives. For example, instead of selecting covariates based on their hypothesized causal relation to the key predictor and outcome of interest [1,9,10,11,12,13,14] , a researcher might include a more limited set of covariates based on common practice within the field or what the researcher finds most interesting. The researcher might address a skeptical reviewer’s concerns by explaining in the discussion section that no causal conclusions can be drawn from nonexperimental studies although the value of the study lies in its contribution towards understanding causal mechanisms. In addition, the researcher might explain that they chose not to consider alternative models because the study’s goals were predictive and not causal. In this process, the researcher would have failed to rule out plausible alternative models. Consequently, the study might avoid the stringent standards of causal evidence to which it would have been subjected if it had explicitly made causal claims.” (p.6-7)

We also explain how authors might use both of these strategies in the second paragraph on page 8 as shown below: 

Additional ways psychologists may implicitly communicate a preferred causal interpretation while hedging to avoid criticism for making causal claims based on correlational data include 1) designing studies for which results are far more interesting or important if they are causally informative and 2) focusing more on their preferred causal interpretation than on ruling out plausible alternatives. For example, consider a hypothetical study that regresses children’s achievement test scores on previous participation in a program designed to improve parent-child interactions, along with a rich set of baseline measures of child and parent skills and contexts, estimating a relation between program participation and adjusted achievement test scores of 0.10 SD. This finding would not be very useful for predictive purposes, nor would it be a useful way to model the process through which some families came to participate in the program and others did not (because test scores are measured after program participation, and thus may be caused by participation). However, if the findings are interpreted as informing readers about the effects of program participation on achievement test scores, a causal interpretation, they are interesting and policy relevant. Further, such recommendations are often given in ways that imply stronger evidence for the relevant causal pathway than warranted by the method (Haber et al., 2022), consistent with the hypothesis that the use of causally ambiguous statistical language may be an example of “strategic ambiguity” (Frankenhuis, Panchanathan, & Smaldino, 2022; Rohrer, 2021). Moreover, the method used may focus more on the author’s preferred causal interpretation than on ruling out plausible alternatives, for example by failing to probe the robustness of results to the inclusion of child prior test scores in the model. For these reasons, avoiding the use of causal language completely is unlikely to prevent readers from treating results as if they are intended to be interpreted as causally informative (because of point 1 above), yet the most plausible set of causal estimates will frequently go unreported (because of point 2). (p.8-9)

Reviewer #2: This manuscript provides an interesting consideration of causal wording and the implications for a readers’ interpretation of the study quality. The study is well designed, and the analyses are appropriate. My main concerns are the proportion of participants who did not pass the manipulation question and the use of “predict” as the non-causal word choice in the sample abstracts.

I found the discussion of predictive models in the introduction section a bit inconsistent and therefore confusing. As the authors’ note in the introduction (no line numbers were included in manuscript), if the purpose of the model is explicitly predictive, then the objective is not to imply an underlying causal mechanism. With this, I agree. However, in several places in the introduction, it is implied that predictive models should be included in a discussion causal wording, presumably because those producing predictive models might use them for intervention purposes? Perhaps this goes back to a need for authors of non-experimental studies to be explicit as to the objective of their study.

 If it is to predict an outcome or to (just) identify associations for further study, then causation is not an aim and it is not necessary to control for confounding (although, the misuse of causal language may still be an issue for these studies). Based on this, I think it is unfortunate that the authors’ chose to use “predict” as their non-causal word in the abstracts evaluated by participants. Is it possible that some of the differences in perceived quality are based on the reader’s understanding of the study purpose (predictive versus causal inference), rather that the causal versus non-causal wording? 

The reviewer raises two interesting points. First, studies that have the primary purpose of prediction for prediction’s sake might not need to discuss their causal assumptions or provide a straightforward causal interpretation of their key estimates. We agree with this point, as stated in the introduction. However, in much psychological research and in the abstracts we used as stimuli, the data are analyzed as if the authors are making some effort to obtain a causal estimate (e.g., by statistically controlling for plausible confounders), results are discussed as if they are causally informative, and no use cases for statistical prediction are given (e.g., a range of scores that could be used as a diagnostic cutoff for some useful clinical purpose). In these cases, we think the author and readers are likely to be interested in causal estimates and discussing assumptions and interpretations of the key coefficients are good practices.

The reviewer also mentions the possibility that a reason many participants failed the manipulation check is that some viewed the study with “predictive” wording as causal and failed the manipulation check whereas others viewed the study as predictive for prediction’s sake and did not. On one hand, we do attribute the poor performance on the manipulation check to the causal ambiguity of the term “predict”. On the other hand, as can be seen from comparing the estimates from Models 3 and 4 in Table 5, readers who passed the manipulation check were slightly MORE likely to endorse the statement that the predictive study supported the policy conclusions warranting causal evidence. Thus, the idea that these participants were merely viewing the predictive study as not causally relevant seems unlikely.

Sentence at end of page 4 starting with “Overall, impeding progress ….” Is an incomplete sentence. In the next sentence, “though” should be “although”.

Thank you for this we have changed the sentence at the end of page 4 to say :

“Although disregarding the causal informativeness of study’s design and analysis can reflect a lack of training in causal inference methods, it may also perpetuate a status quo in which the norm of using causally ambiguous statistical language precludes progress toward more causally informative research in nonexperimental psychology.” (p.4-5)

To me, one of the more interesting findings of this manuscript was the proportion of individuals who erred on the manipulation question was approximately half. This was only superficially discussed but could relate to a general lack of understanding of what words have causal implications. It also may relate to my previous comment on the choice to use “predicts” as the non-causal wording in the abstracts, rather than “associated with” or some other phrase that isn’t associated with a different type of study objective (like predictive modelling). The authors did consider the implications by conducting different model specifications using (as one model) only participants passing this question. Nonetheless, the issue of word choice and the question of why so many participants did not pass the manipulation question could be more comprehensively discussed.

We completely agree and have included these points in the limitations section of the manuscript as shown below: 

“Second, we only tested two different types of abstracts with the term “predict” reflecting the causally ambiguous condition. A potential issue with using the term “predict” is that individuals may have interpreted it as reflecting a study exclusively concerned with predictive modelling. This likely explains the high error rate on our manipulation check. Future work might benefit from using words such as “associate” and “link” which Haber and colleagues [17] identified as being more common and causally ambiguous than the term “predict” – although at least half of reviewers rated all three of these words as suggesting a causal link. Whether this would improve nonexperimental research on net would depend on the extent to which these words increase perceived study quality yet continue to convey some degree of causal information to the reader. Finally, perhaps most important for guiding best practices related to causal language is the extent to which wording affects the author’s thinking about causal assumptions and interpretations, which we hope future work will measure.” (p.32-33)

---

## [Decision Letter · Decision Letter 2]

12 Mar 2023

PONE-D-22-24784R2Hedges, mottes, and baileys: Causally ambiguous statistical language can increase perceived study quality and policy relevancePLOS ONE

Dear Dr. Alvarez-Vargas,

Thank you for submitting your manuscript to PLOS ONE. After careful consideration, we feel that it has merit but does not fully meet PLOS ONE’s publication criteria as it currently stands. Therefore, we invite you to submit a revised version of the manuscript that addresses the points raised during the review process.

We look forward to receiving your revised manuscript.

Kind regards,

Diego A. Forero, MD; PhD

Academic Editor

PLOS ONE

Journal Requirements:

Reviewers' comments:

Reviewer's Responses to Questions

**Comments to the Author**

1. If the authors have adequately addressed your comments raised in a previous round of review and you feel that this manuscript is now acceptable for publication, you may indicate that here to bypass the “Comments to the Author” section, enter your conflict of interest statement in the “Confidential to Editor” section, and submit your "Accept" recommendation.

Reviewer #1: (No Response)

Reviewer #3: (No Response)

2. Is the manuscript technically sound, and do the data support the conclusions?

Reviewer #1: (No Response)

Reviewer #3: Yes

3. Has the statistical analysis been performed appropriately and rigorously? 

Reviewer #1: (No Response)

Reviewer #3: Yes

4. Have the authors made all data underlying the findings in their manuscript fully available?

Reviewer #1: (No Response)

Reviewer #3: Yes

5. Is the manuscript presented in an intelligible fashion and written in standard English?

Reviewer #1: (No Response)

Reviewer #3: Yes

6. Review Comments to the Author

Reviewer #1: I would like to thank the authors for adequately addressing the point that I raised in my last review.

One last thing: I think I noticed two typos in one sentence on page 13. On page 13, the authors stated "a week to moderately root word for implying causation". I guess it should be written instead: "a weak to moderate root word for implying causation".

best,

Michael P. Grosz

Reviewer #3: (No Response)

7. PLOS authors have the option to publish the peer review history of their article (what does this mean?). If published, this will include your full peer review and any attached files.

Reviewer #1: **Yes: **Michael P. Grosz

Reviewer #3: No

---

## [Author Response · Author response to Decision Letter 2]

11 Apr 2023

Reviewer #1:

 I would like to thank the authors for adequately addressing the point that I raised in my last review.

One last thing: I think I noticed two typos in one sentence on page 13. On page 13, the authors stated "a week to moderately root word for implying causation". I guess it should be written instead: "a weak to moderate root word for implying causation".

best,

Michael P. Grosz

Thank you for your careful attention to our manuscript. We have changed the sentence on page 13 and it now reads: “We suspect the latter explanation may be correct (indeed, a recent study reported that about half of respondents view the word “predict” as a weak to moderate root word for implying causation [17]), and thus this manipulation check was not a strong measure of participant attention.”

Reviewer #3

Dear editor,

Thank you for sending the paper titled, ‘Hedges, mottes, and baileys: Causally ambiguous statistical language can increase perceived study quality and policy relevance,’ for me to review. In this paper, the authors provide an overview of the use of causal language in observational studies, focusing on the examples from psychology. They also conduct a survey based study to assess how causal vs. more statistical language influences readers’ interpretation of study results and find that more ambiguous statistical language leads readers to rank papers as higher quality. The authors appropriately note that the common practice of using ambiguous language may be intended as a cautionary step by authors to avoid over interpretation of results. However, in cases in which explicit hypotheses are being tested, the use of non-causal language may instead lead both authors and readers to overlook important assumptions and sources of bias, without imparting a more cautious interpretation of the results or policy recommendations. Overall, I found the paper topic to be both important and timely. Moreover, I agree with previous reviewer’s comments and was glad to see that the authors addressed these issues. In addition to these broad sentiments, below I pose a few questions and provide some (minor) suggestions that I hope improve the manuscript. Finally, I note that I am not a psychologist, so I refrained from commenting on discipline specific content, e.g., how psychology research informs public policy. 

Questions:

1. In the introduction section, ‘Potential costs and benefits of a norm against causal language,’ the authors present the argument that despite research that is often based on testing hypotheses, authors often avoid the use of causal methods and causal language, while still making interpretations and recommendations in the discussion of their papers that are based on causal relationships. I thought the authors provide an even handed discussion here, and I agree that a mismatch between the study question, the methods, the language in how results are presented, and the interpretation of research/policy recommendation is problematic. This section could, in my opinion, be even more direct in emphasizing the importance of clear study question that matches the analytical approach. The author’s might recommend that papers identify the data analysis task upfront (i.e., description, prediction, association, or causal inference), as a way to guide the analysis and reduce ambiguity in the interpretation of results (Cf. https://doi.org/10.1080/09332480.2019.1579578;
https://doi.org/10.1098/rspb.2020.2815;
https://doi.org/10.1002/ecy.3336). In summary, I think the introduction would be more impactful if the authors explicitly identify and create examples based on a specific data analysis task. From here, they can describe how results can be confused by ambiguous language that does not reflect the study question/data analysis task (e.g., a causal model is built but causal language and key assumptions are excluded from the text, or a predictive model is built and causal implications are made in the text). Currently, it is not always clear what data analysis task applies to the example, which makes it harder for the reader to know how they should interpret the results.

We really appreciate this recommendation and have reformatted the introduction section to include these examples as you described. Please find the revised section beginning in page 5 shown below: 

A possible unintended consequence of the norm, however, can occur when the authors research question and analytic approach does not match the interpretation of the implications of the results. For example, if an author’s stated interest is in determining whether children’s early academic skills have an “effect” on later academic achievement, but they conduct an analysis that only includes two variables, there would be a mismatch between the stated goal and the analytical approach because it does not rule out plausible alternatives, therefore likely producing biased estimates and interpretations due to potential confounders. This analytic approach would upwardly bias the estimated effect of children’s early academic skills if instead of selecting covariates based on their hypothesized causal relation to the key explanatory variable and independent variable of interest [1,4,5,6,7,8,9], a researcher includes a more limited set of covariates based on common practice within the field. A separate problem may arise if a researcher overly-adjusts their model by adding in variables that are simply of interest to them. If one of these variables is a potential mediator, then the estimates may be downwardly biased in an over-adjusted model. If the researcher then uses causally ambiguous statistical language to explain their results from these alternative models, for example concluding that “children’s early academic skills influence later academic achievement, which has important policy implications”; this may permit studies that seem causal to be published without being held to the same standards of evidence as studies that make causal claims explicit [10]. The researcher might address a skeptical reviewer’s concerns by explaining in the discussion section that no causal conclusions can be drawn from nonexperimental studies although the value of the study lies in its contribution towards understanding causal mechanisms. In addition, the researcher might explain that they chose not to consider alternative models because the study’s goals were predictive and not causal. In this process, the researcher would have failed to rule out plausible alternative models. Consequently, the study might avoid the stringent standards of causal evidence to which it would have been subjected if it had explicitly made causal claims.

In contrast an author that transparently states their interest in estimating the causal effect of children’s early academic skills on their later academic achievement, it will be clear to an informed reader that an analytical approach with no adjustment for potential confounders, including a variety of explanatory cognitive, emotional, and contextual variables, will be vulnerable to omitted variables bias, likely in the upward direction) . Models that attempt to adjust for such confounds might be subject to further robustness checks or falsification tests that could detect the presence of and perhaps establish the direction and range of plausible magnitudes of additional bias in these estimates [11,12,13]. These models might even make quantitative predictions that could be tested in a subsequent experiment [14]. This is why it is recommended that authors identify their research goals (i.e., description, prediction, association, or causal inference) and analytic plans (i.e., predictive model, causal model) to reduce the ambiguity in the interpretation of the results [15,16,17]. 

2. On a related note, I agree with the previous reviewer and the authors that the use of the word prediction/predictor should be used for studies with the explicit goal of prediction. Therefore, depending on the context, I would use a more general word to describe x variables (e.g., explanatory variable, independent variable etc.).

We have replaced the general words with explanatory variable and independent variable as shown in the previous response in bold face. 

3. Discussing different types of bias that can be introduced when failing to adjust or over adjusting in models used for causal inference might further help clarify to the reader why language should reflect the study question and analytical methods. 

This is a great point, we discuss two different types of bias that are introduced when failing to adjust for potential confounders, or over-adjusting models on page 5, which is also quoted in our reply to the reviewer’s point 1, above and highlighted in bold face. 

I think empirical part of this study uses a clever experimental design. My apologies if I overlooked this, but were the treatments random with respect to the background characteristics…both in the total sample (n=142) and the reduced sample (n=55)? 

You did not miss this; it is a great suggestion. We have included an equivalence table in the Supplementary Material as Table S10 (also shown below). There is only some indication of non-random assignment that took place by condition for the full sample, and some individuals were excluded from the pre-registered sample because they did not meet the exclusion criteria.

Table S10

Demographic Variable Equivalence Across Full Sample and Preregistered Sample 

 Full Sample Pre-registered Sample

 M F p M F p

PsychMD 0.02 0.12 0.73 0.01 0.03 0.87

cogdevsoc listserv 0.07 0.28 0.60 0.29 1.15 0.29

Male 0.13 0.63 0.43 0.14 0.56 0.46

Female 0.00 0.00 0.99 0.04 0.16 0.69

Non-binary /Transgender /Preferred not to answer 0.02 0.37 0.54 0.03 0.79 0.38

Age 3.53 0.04 0.84 4.05 0.05 0.83

PhD Student 0.35 1.55 0.22 0.37 1.54 0.22

Postdoctoral Researcher 0.01 0.07 0.79 0.04 0.18 0.67

Faculty 0.01 0.05 0.83 0.17 0.73 0.40

Masters / Other 0.00 0.00 0.98 

United States 0.16 0.65 0.42 0.04 0.19 0.66

Canada 0.13 2.01 0.16 0.00 0.01 0.92

All other countries 0.03 0.20 0.65 0.00 0.00 0.98

Native English Speaker 0.06 0.25 0.62 0.20 0.86 0.36

Before Age 6 0.00 0.03 0.86 0.17 2.02 0.16

Between Ages 7-10 0.02 0.17 0.68 0.03 0.19 0.67

Between Ages 11-14 0.13 1.68 0.20 0.04 0.43 0.52

Between Ages 15-18 0.01 1.82 0.18 

Statistical Courses Taken 1.02 0.14 0.71 0.53 0.05 0.82

Psychology 0.01 0.10 0.76 0.01 0.20 0.66

Other Field 0.39 4.73 0.03 * 0.01 0.20 0.66

Clinical 0.20 1.97 0.16 0.04 2.05 0.16

Cognitive 0.02 0.11 0.74 0.31 1.47 0.23

Neuroscience 0.00 0.01 0.92 0.00 0.05 0.83

Developmental 0.05 0.22 0.64 0.08 0.31 0.58

Quantitative 0.01 0.16 0.69 0.13 1.13 0.29

Social Personality 0.06 0.47 0.49 0.00 0.02 0.89

Educational 0.12 1.66 0.20 0.20 2.06 0.16

Experimental 0.01 0.42 0.52 

Forensic 0.01 1.76 0.19 0.02 1.29 0.26

History 0.01 1.82 0.18 

Industrial 0.00 0.19 0.67 0.00 0.06 0.80

School 0.01 1.82 0.18 0.04 2.31 0.13

Survey Duration 2578029 1.75 0.19 3920048 1.26 0.27

Excluded 0.65 2.73 0.10 

Note. *** p< 0.001, **p<0.01, *p< 0.05. Each estimate was derived from an One way ANOVA in which the demographic variable was regressed on the Condition variable separately for the full sample and for the pre-registered sample. PsychMD = Psychological Methods Discussion Facebook group. F is ANOVA F-statistic for the condition variable with 4 categories. Descriptive statistics of full sample and restricted pre-registered analytic sample after cases that met exclusionary criteria were removed. Participants that preferred not to answer questions or did not answer were reported together to prevent any possibility of identification, such as by grouping non-binary and transgender groupings as well as countries of employment. Missing rows indicate that individuals with those demographic variables were not present in the sample. 

4. This is merely a note, but I found the discussion and the warnings/recommendations therein to be well written and something every-reviewer/author should consider…i.e., the use of causal language, when assumptions are explicit and sources of bias recognized, is much more useful than sweeping those details under the veil of statistically ambiguous language and proceeding boldly with the inference/policy recommendations. 

This is a very important note, we have referenced your statement and included it in the manuscript at the bottom of page 30 as shown below. If the reviewer would like, we’d be happy to reword it in our own language.

“Merely replacing causal language with causally ambiguous statistical language is no substitute for these difficult but essential steps. Researchers should consider using causal language, when the assumptions of an analysis are explicit, and sources of bias recognized. This is much more useful scientific endeavors than sweeping those details under the veil of statistically ambiguous language and proceeding boldly with the inference/policy recommendations[4]. We encourage future research on how causal language can best be employed to communicate researchers’ communication about psychological research in ways that address the underlying theory under consideration but do not convey overconfidence about causal implications. 

[4] We thank an anonymous reviewer for making this point.”

Minor Comments:

1. Pg 5, last paragraph. A fewer number of variables and the use of a single model does not distinguish a predictive model from a causal model (e.g., predictive models often perform model selection to identify the set of variables that explain the most variation in an outcome…this process often involves multiple models and possibly many variables since a predictive model is not concerned about relationships among the independent variables). Related to my first comment in the ‘questions’ section, I wonder if clarifying the examples and then highlighting how they can be mis-interpreted causally, would be helpful. 

2. Pg 7, first paragraph. I would disagree that causal inference is ‘much more attractive’ than prediction. They are simply different analytical tasks to answer different questions. 

We have made changes here but reread it to make sure it is comprehensible. 

3. Pgs 8-9. This exact sentence is repeated on these pages, “such recommendations are often given in ways that imply stronger evidence for the relevant causal pathway than warranted by the method [17] consistent with the hypothesis that the use of causally ambiguous statistical language may be an example of “strategic ambiguity” [20, 22].”

We have removed the duplicate sentence on page and kept the sentence in page 9. Thank you for your careful attention to this. 

4. Pg 9, last sentence. Why does a more complicated model result in a smaller estimate? 

We have replaced the last sentence to explain a scenario in which an estimate may be reduced by testing causal assumptions on page 9 as follows: 

“In addition, contrasting estimates from increasingly well-controlled models to account for confounding variables or conduct sensitivity and robustness tests often yields smaller associations between the key dependent and independent variables. When this is the case, a straightforward causal framing of the reduced estimates may be detrimental for the perceived quality and policy-relevance of one’s work.”

5. Pg 10. What does the effect size of 0.30 represent…a difference in some unitless score?

We have revised this section to explain the relative magnitude of this effect size in the context of our survey. The text now reads as follows on the bottom of page 12:

We report unstandardized and standardized estimates. For example, in the first column of Table 2, we estimate that participants rated design and study quality .44 points higher on a 5 point Likert scale when abstracts were worded in causally ambiguous statistical language. This effect corresponded to .54 SD impact on the outcome of interest.

6. Pgs 12-13. The responses to the number of individuals who failed the manipulation check are contradictory…is it 79, as noted on pg 12, or 64 as indicated on pg 13. In both cases, these are not 44% of 142.

Thank you for catching this error, the correct number of people that failed the manipulation check is 79. We have fixed this in page 12 and page 13. As shown below:

Many participants failed the manipulation check (55%) which asked: “How many of the two abstracts you just read employed explicitly causal language in their description of the results?” with three responses (0) Neither (n=11), (1) Only one (n=63), (2) Both (n=53), and some participants did not respond (n=15).

7. Pg 13 and 18. Final sample sizes after exclusion criteria (n=55, pg 13) and in the final model (n=58, pg 18) do not agree.

The correct sample size of the final model is 55. We have changed the label on the tables to make this clearer and we have fixed this in page 13 and page 18. Thank you for pointing out this discrepancy. 

8. Table 2, column headers are not consistent…Abstract 1 does not include sample size in header like Abstract 2.

We have fixed this discrepancy.

9. In Tables 4 and 5, the subtitle says that model 1 uses only the first abstracts read (“Model 1 is an ordinary least squares model using only the quality response from the first abstract seen by the participants.”), but in fact based on the model specifications table and the table above that including the estimates, this must be incorrect (e.g., model 2 has no estimate for order).

This is correct, we made a mistake in updating this note, and have fixed it to say that it is indeed model 2 that is “using only the quality response from the first abstract seen by the participants”. See full revised notes below: 

Note. *** p< 0.001, **p<0.01, *p< 0.05. Although our full sample size was 142, 8 cases were missing the second abstract and were excluded from the analyses.

PsychMD = Psychological Methods Discussion Facebook group. The dependent variable is Question 1:“Based on the limited information available in this abstract, how would you rate the quality of the design and analysis of the study described? “is answered on a scale from 1 – 5 where 5 =Very high quality, 4 = High quality, 3 = Moderate quality, 2 = Low quality, and 1= Very low quality. Causally Ambiguous Statistical Language is a dummy variable in which causal language used in the abstract = 0 and Causally Ambiguous Statistical Language = 1. The abstract variable is coded as 1 = abstract 1 and 2 = abstract 2 to indicate effects of abstract used on quality rating. Order indicates which abstract was seen first (1) or second (2) to measure the effects of abstract order on quality rating. Model 2 is an ordinary least squares model using only the quality response from the first abstract seen by the participants. All other models use individual random effects to estimate the effect of the independent variables on changes in quality rating within-person since all participants saw to abstracts and provided two quality ratings. SD indicates one standard deviation for the outcome variable. An effect size can be calculated by dividing regression estimates by the outcome standard deviation. R2 reflects the pseudo r-squared for fixed effect. Model 4 represents the preregistered sample, 84 cases were excluded because participants answered first two questions in under 30 seconds (n=12), participants were not in a field related to psychology (n=6), they failed the manipulation check (n=64), were not enrolled in a PhD program, post-doctoral position, or a faculty position(n = 1), or they were not at least 18 years-old (n=1).

10. Figure 1 caption: subject verb agreement issue, “abstracts that have”

We have fixed this grammatical error, thank you!

---

## [Decision Letter · Decision Letter 3]

26 Apr 2023

PONE-D-22-24784R3Hedges, mottes, and baileys: Causally ambiguous statistical language can increase perceived study quality and policy relevancePLOS ONE

Dear Dr. Alvarez-Vargas,

Thank you for submitting your manuscript to PLOS ONE. After careful consideration, we feel that it has merit but does not fully meet PLOS ONE’s publication criteria as it currently stands. Therefore, we invite you to submit a revised version of the manuscript that addresses the points raised during the review process.

We look forward to receiving your revised manuscript.

Kind regards,

Diego A. Forero, MD; PhD

Academic Editor

PLOS ONE

Journal Requirements:

Reviewers' comments:

Reviewer's Responses to Questions

**Comments to the Author**

1. If the authors have adequately addressed your comments raised in a previous round of review and you feel that this manuscript is now acceptable for publication, you may indicate that here to bypass the “Comments to the Author” section, enter your conflict of interest statement in the “Confidential to Editor” section, and submit your "Accept" recommendation.

Reviewer #3: (No Response)

2. Is the manuscript technically sound, and do the data support the conclusions?

Reviewer #3: Yes

3. Has the statistical analysis been performed appropriately and rigorously? 

Reviewer #3: Yes

4. Have the authors made all data underlying the findings in their manuscript fully available?

Reviewer #3: Yes

5. Is the manuscript presented in an intelligible fashion and written in standard English?

Reviewer #3: Yes

6. Review Comments to the Author

Reviewer #3: Dear editor,

I have read through the authors' point-by-point response and the revisions that they made to the manuscript titled, ‘Hedges, mottes, and baileys: Causally ambiguous statistical language can increase perceived study quality and policy relevance’. Upon reading through point-by-point, I found that the authors did a nice job of addressing my prior suggestions. Here, I provide a few final suggestions for their consideration.

1. In the introduction, starting on page 5, the first sentence the authors added in the revised draft seems a bit unclear to me. When they state, “…analytic approach does not match the interpretation of the implications of the results.” I am no sure what is meant by implications…maybe policy implications, or maybe the word implications can be omitted altogether.

2. A few lines down on from the previously mentioned sentence on page five, ‘two variables are discussed’ in the context of a simple regression model. More specifically, they state “…upwardly bias the estimated effect of children’s early academic skills if instead of selecting covariates based on their hypothesized causal relation to the key explanatory variable and independent variable of interest…” I think the authors might mean dependent variable of interest, since confounding variables are both associated with the explanatory variable and the outcome.

3. Regarding question 4 from my previous review, the authors asked if they needed to reword their revision…I have no problem with them incorporating text from my review into their manuscript.

4. In my previous minor revision #4, the authors response in which they note that adjustment for confounds may lead to a smaller estimate of interest, they then say that “…a straightforward causal framing of the reduced estimates may be detrimental…” I think this sentence could be clearer. The reduced size of the estimate may lead some policy makers to perceive the study as less conclusive, but the explanation of causal inference methods per se should speak to the robustness of the relationship between x and y, since causal inference can help explain why estimates change from one model to the next.

7. PLOS authors have the option to publish the peer review history of their article (what does this mean?). If published, this will include your full peer review and any attached files.

Reviewer #3: No

---

## [Author Response · Author response to Decision Letter 3]

11 May 2023

Reviewer #3: 

Dear editor,

I have read through the authors' point-by-point response and the revisions that they made to the manuscript titled, ‘Hedges, mottes, and baileys: Causally ambiguous statistical language can increase perceived study quality and policy relevance’. Upon reading through point-by-point, I found that the authors did a nice job of addressing my prior suggestions. Here, I provide a few final suggestions for their consideration. 

1. In the introduction, starting on page 5, the first sentence the authors added in the revised draft seems a bit unclear to me. When they state, “...analytic approach does not match the interpretation of the implications of the results.” I am no sure what is meant by implications...maybe policy implications, or maybe the word implications can be omitted altogether. 

We have revised this sentence to clarify that we are discussing the policy implications. The first sentence on page 5 now reads: 

A possible unintended consequence of the norm, however, can occur when the authors research questions and analytic approaches do not match the interpretation of the policy implications of the results.

2. A few lines down on from the previously mentioned sentence on page five, ‘two variables are discussed’ in the context of a simple regression model. More specifically, they state “...upwardly bias the estimated effect of children’s early academic skills if instead of selecting covariates based on their hypothesized causal relation to the key explanatory variable and independent variable of interest...” I think the authors might mean dependent variable of interest, since confounding variables are both associated with the explanatory variable and the outcome. 

We have corrected this sentence and it now reads as follows: This analytic approach would upwardly bias the estimated effect of children’s early academic skills if instead of selecting covariates based on their hypothesized causal relation to the key explanatory variable and dependent variable of interest [1,4,5,6,7,8,9], a researcher includes a more limited set of covariates based on common practice within the field.

3. Regarding question 4 from my previous review, the authors asked if they needed to reword their revision...I have no problem with them incorporating text from my review into their manuscript.

Thank you!

4. In my previous minor revision #4, the authors response in which they note that adjustment for confounds may lead to a smaller estimate of interest, they then say that “...a straightforward causal framing of the reduced estimates may be detrimental...” I think this sentence could be clearer. The reduced size of the estimate may lead some policy makers to perceive the study as less conclusive, but the explanation of causal inference methods per se should speak to the robustness of the relationship between x and y, since causal inference can help explain why estimates change from one model to the next. 

This is a great point for clarification; we have modified the sentence on page 10 as shown below: 

In addition, contrasting estimates from increasingly well-controlled models to account for confounding variables or conduct sensitivity and robustness tests often yields smaller associations between the key dependent and independent variables. When this is the case, a straightforward causal framing of the reduced estimates speaks to the robustness of the relationship between the key dependent and independent variables. However, it may be detrimental for the perceived quality and policy-relevance of one’s work if policy makers perceive the study to be less conclusive.

---

## [Editor Report · Decision Letter 4]

16 May 2023

Hedges, mottes, and baileys: Causally ambiguous statistical language can increase perceived study quality and policy relevance

PONE-D-22-24784R4

Dear Dr. Alvarez-Vargas,

We’re pleased to inform you that your manuscript has been judged scientifically suitable for publication and will be formally accepted for publication once it meets all outstanding technical requirements.

Kind regards,

Diego A. Forero, MD; PhD

Academic Editor

PLOS ONE
---

## [Editor Report · Acceptance letter]

21 Jun 2023

PONE-D-22-24784R4 

Hedges, mottes, and baileys:
Causally ambiguous statistical language can increase perceived study quality and policy relevance 

Dear Dr. Alvarez-Vargas:

I'm pleased to inform you that your manuscript has been deemed suitable for publication in PLOS ONE. Congratulations! Your manuscript is now with our production department. 

Kind regards, 

on behalf of

Dr. Diego A. Forero 

Academic Editor

PLOS ONE